Manuscript prepared for Hydrol. Earth Syst. Sci.
with version 2015/04/24 7.83 Copernicus papers of the LaTeX class copernicus.cls.
Date: 15 June 2018

# Exploring the merging of the global land evaporation WACMOS-ET products based on local tower measurements

Carlos Jiménez[1,2], Brecht Martens[3], Diego M. Miralles[3], Joshua B. Fisher[4], Hylke E. Beck[5], and Diego Fernández-Prieto[6]

[1]Estellus, Paris, France
[2]LERMA, Paris Observatory, Paris, France
[3]Laboratory of Hydrology and Water Management, Ghent University, Ghent, Belgium
[4]Jet Propulsion Laboratory, California Institute of Technology, Pasadena, California, USA
[5]Department of Civil and Environmental Engineering, Princeton University, Princeton, New Jersey, USA
[6]ESRIN, European Space Agency, Frascati, Italy

*Correspondence to:* Carlos Jiménez(carlos.jimenez@estellus.fr)

**Abstract.** An inverse error variance weighting of the anomalies of three terrestrial evaporation (ET) products from the WACMOS-ET project based on FLUXNET sites is presented. The three ET models were run daily and at a resolution of 25 km for 2002–2007, and based on common input data when possible. The local weights, derived based on the variance of the difference between the tower ET anomalies and the modelled ET anomalies, were made dynamic by estimating them using a 61-day running window centered on each day. These were then extrapolated from the tower locations to the global landscape by regressing them on the main model inputs and derived ET using a neural network. Over the stations, the weighted scheme usefully decreased the random error component, and the weighted ET correlated better with the tower data than a simple average. The global extrapolation produced weights displaying strong seasonal and geographical patterns, which translated into spatiotemporal differences between the ET weighted and simple average ET products. However, the uncertainty of the weights after the extrapolation remained large. Out-sample prediction tests showed that the tower data set, mostly located at temperate regions, had limitations with respect to the representation of different biome and climate conditions. Therefore, even if the local weighting was successful, the extrapolation to a global scale resulted problematic, showing a limited added value over the simple average. Overall, this study suggests that merging tower observations and ET products at the time and spatial scales of this study is complicated by the tower spatial representativeness, the products coarse spatial resolution, the nature of the error in both towers and gridded data sets, and how all these factors impact the weights extrapolation from the tower locations to the global landscape.

## 1 Introduction

The surface latent heat flux governs the interactions between the Earth and its atmosphere (Betts, 2009), is an essential component of the water and energy cycles (Sorooshian et al., 2005), and thus plays a key role in the climate system and on the linking of biochemical cycles (Wang and Dickinson, 2012). Terrestrial evaporation (ET) – the associated flux of water from land into the atmosphere – is also an important variable in the management of agricultural systems, forests, and hydrological resources. Hence, estimates of ET at different spatial scales, ranging from individual plants for managing irrigation, to basin scales to evaluate water availability, are required by many applications(e.g. Dunn and Mackay, 1995; Le Maitre and Versfeld, 1997; Gowda et al., 2008; Fisher et al., 2017).

Point-based measurements of land heat fluxes are typically conducted during field experiments (Pauwels et al., 2008) or by more permanent monitoring systems, such as lysimeters (Hirschi et al., 2017) and flux tower networks (Baldocchi et al., 2001). However, these are ultimately point measurements that require specific equipment and cannot be applied for routine monitoring over large areas. Therefore, more readily available meteorological observations are often combined with well known flux formulations (e.g., Monteith, 1965; Priestley and Taylor, 1972) to obtain regional-scale estimates.

To derive global estimates, a central challenge remains: ET does not have a direct signature that can be remotely detected. As an alternative, satellite remote sensing observations related to surface temperature, soil moisture, or vegetation can again be combined with traditional flux formulations (e.g., Monteith, 1965; Priestley and Taylor, 1972) to derive global estimates at different time and spatial scales. This has led to the raise and proliferation of satellite observation-based retrieval models (and subsequent data sets) of ET over the last few years (for overview see Wang and Dickinson, 2012; Zhang et al., 2016). Global flux estimates are also available from atmospheric reanalyses (e.g. Dee et al., 2011), but are often treated separately as they are not as directly constrained by observations as the satellite data-driven data sets (Jimenez et al., 2011; Mueller et al., 2013). In addition, the latter are specifically designed to estimate ET, and while also uncertain, their errors are in principle more traceable due to their lower complexity. Nonetheless, satellite-based ET products also show large discrepancies which are put in evidence when inter-compared and evaluated against in situ flux networks (Jimenez et al., 2011; Mueller et al., 2011; McCabe et al., 2016).

Far from discouraging the use of these ET datasets, the inter-product differences have been perceived as an opportunity to foster research and find new means to combine these datasets in an optimal manner. So far, these efforts have ranged from simply averaging a number of ET products (Mueller et al., 2013) to more complex approaches, such as weighted averages (Hobeichi et al., 2018), fusion algorithms where the original ET products are combined to reproduce flux observations (Yao et al., 2017), or integration methodologies that seek consistency between ET products and related products of the water cycle (Aires, 2014; Munier and Pan, 2014). ET products based on a direct regression of tower ET on a set of explanatory variables also exist(Jung et al., 2011).

Aiming at improving the predictive capability for ET, the WAter Cycle Multi-mission Observation Strategy – ET project (WACMOS-ET, *http://wacmoset.estellus.eu*) compiled a forcing data set covering the period 2005–2007, and ran four established ET models using common forcing to explore the uncertainties and accuracy of the underlying algorithms (Michel et al., 2016; Miralles et al., 2016). Three of the models – the Priestley-Taylor Jet Propulsion Laboratory model (PT-JPL, Fisher et al., 2008), the Global Land Evaporation Amsterdam Model (GLEAM, Miralles et al., 2016), and the Penman–Monteith algorithm from the MODerate resolution Imaging Spectroradiometer (MODIS) evaporation product (PM-MOD, Mu et al., 2011) – were run to produce 3-hourly and daily estimates at $0.25^o$ spatial resolution. As far as we know, they remain the only publicly available global ET estimates at these spatiotemporal resolutions.

Analyses of the WACMOS-ET estimates showed substantial differences between the three model products, both at the point scale (Michel et al., 2016) as well as globally (Miralles et al., 2016). As such we here pose the question: can a combination of these estimates result in accurate ET? The simplest approach is to assume that all products are equally uncertain, merging them with a simple average. A more elaborated approach is to assign weights to each product based on an accurate description of the specific product uncertainties. However, even if some attempts to derive model uncertainty exist (Miralles et al., 2011a; Badgley et al., 2015; Loew et al., 2016), the complexity to derive estimates of ET from remote sensing data means that reliable quality assessment is only attained through validation against tower flux measurements. Therefore, here we explore a local flux tower-based weighting of GLEAM, PT-JPL, and PM-MOD and compare it with the more typical simple average, followed by an appraisal of the potential to globally extrapolate the resulting merging framework.

## 2 Methods

### 2.1 ET models

The GLEAM, PT-JPL, and PM-MOD models, and the inputs required to run them globally at a $0.25^o$ spatial resolution are extensively described by Michel et al. (2016) and Miralles et al. (2016). Only the main differences with respect to the original WACMOS-ET runs are fully detailed here. Note that the original 2005-2007 period is extended here to cover 2002-2007, and that the models are only run at daily time resolutions.

#### 2.1.1 GLEAM

GLEAM is a simple land surface model fully dedicated to deriving evaporation. It distinguishes between direct soil evaporation, transpiration from short and tall vegetation, snow sublimation, open-water evaporation, and interception loss from tall vegetation. Interception loss is independently calculated based on the Gash (1979) analytical model forced by observations of precipitation. The re-

maining components of evaporation are based upon the formulation by Priestley and Taylor (1972) for potential evaporation, constrained by multiplicative stress factors. For transpiration and soil evaporation, the stress factor is calculated based on the content of water in vegetation (microwave vegetation optical depth) and the root zone (multilayer soil model driven by observations of precipitation and updated through assimilation of microwave surface soil moisture). For regions covered by ice and snow, sublimation is calculated using a Priestley and Taylor equation with specific parameters for ice and supercooled waters. For the fraction of open water at each grid cell, the model assumes potential evaporation.

The recent GLEAM v3 model of Martens et al. (2016) is adopted here and replaces the model of Miralles et al. (2011) previously applied for the WACMOS-ET runs. Major differences related to the previous model are a revised formulation of the evaporative stress, an optimized drainage algorithm, and a new soil moisture data assimilation system.

### 2.1.2   PT-JPL

The PT-JPL model by Fisher et al. (2008) is a relatively simple algorithm to derive ET. It uses the Priestley and Taylor (1972) approach to estimate potential evaporation, and then applies a series of stress factors to reduce from potential to actual evaporation. The land evaporation is partitioned first into soil evaporation, transpiration, and interception loss by distributing the net radiation to the soil and vegetation components. Unlike GLEAM, the stress factors in PT-JPL are based on atmospheric moisture (vapour pressure deficit and relative humidity) and vegetation indices (normalized difference vegetation index, and soil adjusted vegetation index) to constrain the atmospheric demand for water. The partitioning between transpiration and interception loss is done using a threshold based on relative humidity, and is therefore conceptually quite different from the precipitation based calculation in GLEAM. There is no independent estimation of snow sublimation, and the same algorithms are applied for snow-covered areas.

For this study, optimized vegetation products are used as inputs to the model. In WACMOS-ET, the Leaf Area Index (LAI) and Fraction of Absorbed Photosynthetic Active Radiation (FAPAR) products, derived from the Joint Research Centre Two-Stream Inversion (JRC-TIP) package (Pinty et al., 2007, 2011a, b), were converted by a simple biome-dependent calibration to a LAI/FAPAR product consistent with the Moderate Resolution Imaging Spectroradiometer (MODIS) LAI/FAPAR before being used as inputs to the model (Michel et al., 2016). Under the assumptions that the JRC-TIP FAPAR is related to the radiation absorption by the green fraction of the canopy, while the MODIS FAPAR is more related to green and non-green leaf area, a new use of the WACMOS-ET vegetation products is proposed. First, the WACMOS-ET JRC-TIP FAPAR is assumed to be close to an Enhanced Vegetation Index (EVI), and it is scaled by the factor 1.2 to become closer to the FAPAR expected by the model, as in the original PT-JPL equations (Fisher et al., 2008). Second, the WACMOS-ET MODIS-like FAPAR is used as the Fraction of Intercepted Photosynthetic Active

Radiation (FIPAR) expected by the model, which in turn is used by the model as a proxy for the fractional total vegetation cover. Using the original relationships in the model, the fractional total vegetation cover is related to a total (green and non-green) LAI, which is then used to partition the net radiation into their soil and canopy components.

### 2.1.3 PM-MOD

The PM-MOD is based on the Monteith (1965) adaptation of Penman (1948), and the version applied here follows the implementation of Mu et al. (2011). It estimates ET as the sum of interception loss, transpiration, and soil evaporation. Aerodynamic and surface resistances for each component of evaporation are based on extending biome-specific conductance parameters to the canopy scale using vegetation phenology and meteorological data. The surface resistance schemes uses LAI, with further constrains based on air temperature and vapour pressure deficit, avoiding the need of soil moisture and wind speed to parameterize the resistances. Different from GLEAM and PT-JPL, which do not use tower-based calibration, some of the resistance parameters require a biome-based calibration derived from a selection of tower measurements. As for PT-JPL, there is no specific parameterization for snow-covered areas.

The WACMOS-ET LAI/FAPAR products are used with PM-MOD as in Michel et al. (2016), i.e., the model is run with the vegetation products rescaled by a biome-dependent calibration to make them consistent with the expected MODIS values. As the biome-based calibration of PM-MOD was derived with MODIS products, any errors introduced by this simple rescaling can propagate to the PM-MOD estimates and can be responsible for some ET patterns differing from the official use of the Mu et al. (2011)algorithm for the MODIS ET product.

## 2.2 Merging technique

### 2.2.1 Tower weighting

The weights in a merging scheme are typically based on an estimation of some measure of product uncertainty. Here the idea is to estimate the weights proportionally to the agreement between the variations of each ET product and the tower measurements. In order to do so, we propose the following merging scheme:

1. At each tower location, both the different ET products and the tower observations are decomposed into a time series of anomalies and a seasonal climatology as follows:

$$E_m = Ea_m + Ec_m \tag{1}$$

where $E_m$ is the GLEAM (G), PT-JPL (P), PM-MOD (M), and tower observations (O) ET, $Ea_m$ their respective anomalies, and $Em_c$ their respective seasonal climatologies. For the ET products, they are obtained by calculating their respective multi-year (2002–2007) daily

averages. Given the relatively short period, they are further smoothed by applying a 30-day moving average filter. For the towers however, the climatology is estimated over all available site years (even if outside the 2002-2007 period) in order to estimate a climatology that is as robust as possible (note that the obtained climatologies are also further smoothed using the same moving average filter).

2. The product anomalies are weighted as follows:

$$Ea_{WA} = \mathbf{w}^T \mathbf{Ea} \tag{2}$$

where $Ea_{WA}$ is the weighted anomaly, $\mathbf{Ea} = [Ea_G, Ea_P, Ea_M]^T$ is the anomaly vector, and $\mathbf{w} = [w_G, w_P, w_M]^T$ is the weight vector calculated as $\mathbf{w} = (\mathbf{1}^T \mathbf{C} \mathbf{1})^{-1} \mathbf{1}^T \mathbf{C}^{-1}$, with $\mathbf{1} = [1,1,1]^T$ and $\mathbf{C}$ the 3x3 error covariance matrix of the differences $Ea_m - Ea_o$ for each product. We expect the errors to have a seasonal dependence. Hence, in order to estimate the temporal evolution of the weights, they are calculated using a moving window, where the error-covariance at a certain point in time is calculated using all available ET estimates within the time window. The choice of window length is subjective: shorter time windows produce more dynamic weights, but their values are likely to be noisier given the smaller number of samples available to estimate the time series variability. A number of 30 days before and after each calendar day was found to provide a good compromise between the smoothness of weights and the number of samples required, so a 61-day running window is used to calculate the daily weights.

3. The merged product is finally calculated by adding the weighted anomalies to the average of the 3 products climatology:

$$E_{WA} = Ea_{WA} + 1/3 \sum_{m=G,P,M} Ec_m \tag{3}$$

where $E_{WA}$ is the weigthed average merged product (WA-merger). Note that the sum of the weights equals one, and that for equally uncertain anomalies the weight vector becomes $[1/3, 1/3, 1/3]^T$. In that case the weighted product corresponds to the simple average (SA-merger) of the individual products.

### 2.2.2 Weights extrapolation

In order to produce a global weighted product, an extrapolation of the weights from the tower space (i.e., the 84 cells where the towers are located, see Section 3.2) to the entire continental land is needed. The approach chosen to predict the weights outside the tower space is to non-linearly regress the weights based on the main ET model inputs and model ET estimates. For the regression, we use a single neural network (NN) modelling the annual statistical relationship between the weights and their predictors. NNs are broadly used given their capability to approximate non-linear functions,

and are in principle a suitable tool to extrapolate the tower weights. Here it is is used to model the statistical distribution of the weights. However, given that the this error distribution does not only depend on the variables used as predictors in the NN approach, the weights can never be perfectly predicted.

A standard multi-layer perceptrons with a 11 inputs first layer, one hidden layer with 30 neurons and sigmoidal activation functions, and one output layer with 3 neurons and linear activation functions, is used for the regression. Inputs to the NN are the GLEAM, PT-JPL, and PM-MOD ET together with the surface net radiation, the near-surface air temperature, the relative humidity, the soil moisture, the vegetation optical depth, and the project LAI and FAPAR (see Section 3.1). The outputs to be predicted by the NN are the GLEAM, PT-JPL, and PM-MOD weights. The NN initial weights are randomly initialized by the Nguyen-Widrow algorithm (Nguyen and Widrow, 1990), and the final weights assigned by a Marquardt-Levenberg backpropagation algorithm (Hagan and Menhaj, 1994) minimizing a standard sum of square errors (Bishop, 1995b). Note that given the statistical nature of the prediction, the sum of weights can slightly differ from the expected value of one. To assure the sum equalling one, the NN predicted weights are normalized by their sum.

The objective of any NN is to model the general distribution of the data, not the very specific features of the training dataset. The existence of these specific features is unavoidable, as any training dataset is always limited in terms of being a sample of the true distribution. Modelling the specific features is often referred to as "over-fitting". To prevent the latter standard techniques such as early stopping are applied (Bishop, 1995a). In practice this involves monitoring the evolution of the NN error function for an independent validation data set, here constructed by randomly sampling 20% of the original training data set. While this error decreases at the beginning of the training, there is a moment when starts to increase again. This is taken as an indication of the NN starting to over-fit, and the training is haltered.

Preventing over-fitting only assures the right NN model complexity for the conditions sampled in the training data set. In this particular case the limited spatial coverage of the tower stations suggest a poor sampling of the global conditions (see Section 3.2), and further tests are required to see the NN capacity to extrapolate to un-sampled conditions. For this, we will apply out-sample techniques where one tower station is removed from the training data set, followed by assessing the NN performance at the removed station. If the performance is poor, this strongly suggests that the training data set is not robust enough to represent conditions not sampled within this training data set distribution. Note that for the early-stopping technique training and validation subsets contain data from the same stations. So, if the out-sample technique is also applied, the data from the removed station is no longer part of the training nor validation subsets during the cross-validation.

Note that as tower measurements were masked for rainy intervals (see Section 3.2), the interception loss of the modelled ET is not evaluated. Therefore, only the sum of soil evaporation and transpiration is compared with the tower data and weighted. To derive the total ET merged product,

an estimate of interception loss should also be provided, either by (1) assuming that GLEAM, PT-JPL, and PM-MOD interception loss are equally uncertain and adding their average to the weighted soil evaporation and transpiration, or; (2) by adding just one of the individual model interception losses, if there are reasons to believe that the selected one is less uncertain. Here we adopt the first

approach, so the total ET product is the sum of the weighted soil evaporation and transpiration, together with the inter-product interception loss.

## 2.3 Metrics

Agreement with the towers ET is analyzed by calculating the Pearson correlation coefficient (R), the Mean Square Difference (MSD), and the Root Mean Square Difference (RMSD) according to the

240 expressions:

$$R = \frac{N \sum_{i=1}^{N} P_i O_i - \sum_{i=1}^{N} P_i \sum_{i=1}^{N} O_i}{\sqrt{[N \sum_{i=1}^{N} P_i{}^2 - (\sum_{i=1}^{N} P_i)^2]} \sqrt{N \sum_{i=1}^{N} O_i{}^2 - (\sum_{i=1}^{N} O_i)^2}} \tag{4}$$

$$MSD = [\frac{1}{N} \sum_{i=1}^{N} (P_i - O_i)^2] = RMSD^2 \tag{5}$$

where $P$ and $O$ are the model-derived and observed (or a second model-derived) variate, and N is

245 the number of cases. The MSD can be decomposed into a random ($MSD_r$) and systematic ($MSD_s$) component following Willmott (1982) by using the expressions:

$$MSD_r = \frac{1}{N} \sum_{i=1}^{N} (\hat{P}_i - O_i)^2 = RMSD_r{}^2 \tag{6}$$

$$MSD_s = \frac{1}{N} \sum_{i=1}^{N} (P_i - \hat{P}_i)^2 = RMSD_s{}^2 \tag{7}$$

where $\hat{P}_i = a + bO_i$ is the linear least squares regression of $P$ onto $O$, being $a$ and $b$ the regression intercept and slope, respectively. Notice that $MSD = MSD_r + MSD_s$.

Statistics are calculated for the complete study period, or separately for the boreal winter (DJF), spring (MAM), summer (JJA), and autumn (SON). For the correlations, statistical significance is tested by calculating 95% confidence intervals. For the correlation differences, a Fisher Z-transformation

is applied to the correlations, and a Student t-test at a 5% significance level used to test the significance of the difference. The autocorrelation of the daily time series is taken into account by reducing the degrees of freedom using an effective sampling size (De Lannoy and Reichle, 2016; Lievens et al., 2017).

## 3 Data

 ### 3.1 Model inputs

The GLEAM, PT-JPL, and PM-MOD required global inputs remain unchanged with respect to Michel et al. (2016) and Miralles et al. (2016), apart from the precipitation product, and are applied at the same resolution of $0.25^o$. Common inputs to the models are the surface net radiation, coming from the NASA and GEWEX Surface Radiation Budget (SRB, Release 3.1 Stackhouse et al., 2004), and the near-surface air temperature, sourced from the ERA-Interim atmospheric reanalysis (Dee et al., 2011). PT-JPL and PM-MOD also require near-surface air humidity, also derived from ERA-Interim, and the vegetation products discussed in Sections 2.1.2 and 2.1.3. On the other hand, GLEAM requires precipitation, coming from the Multi-Source Weighted-Ensemble Precipitation (MSWEP) version 1 product (Beck et al., 2017), soil moisture and vegetation optical depth from the European Space Agency (ESA) Climate Change Initiative (CCI) Soil Moisture v2.3 product (Liu et al., 2011b, a), and information on snow water equivalents, from the ESA GlobSnow product for the Northern Hemisphere (Takala et al., 2011), and from the National Snow and Ice Data Center (NSIDC) in snow-covered regions of the Southern Hemisphere (Kelly et al., 2003).

**Table 1.** List of the FLUXNET sites used in this study together with their FLUXNET code (ID), IGBP land cover (LC) and official reference or principal investigator (PI). The CA-NS1-7 refers to seven stations closely located and run by the same group.

| ID | LC | Reference/PI | ID | LC | Reference/PI | ID | LC | Reference/PI |
|---|---|---|---|---|---|---|---|---|
| AT-Neu | GRA | George Wohlfahrt | AU-How | SAV | Jason Beringer | BE-Bra | MF | Ivan Janssens |
| BE-Bra | MF | Ivan Janssens | BE-Lon | CRO | Moureaux et al. (2006) | BE-Vie | MF | Aubinet et al. (2001) |
| BR-Sa3 | EBF | Steininger (2004) | CA-Gro | MF | McCaughey et al. (2006) | CA-Man | ENF | Dunn et al. (2007) |
| CA-NS1-7 | ENF | B.Lamberty et al. (2004) | CA-Oas | MF | Bond-Lamberty et al. (2004) | CA-Obs | ENF | Bond-Lamberty et al. (2004) |
| CA-Qfo | ENF | Bergeron et al. (2007) | CA-SF1 | ENF | Coursolle et al. (2012) | CA-SF2 | MF | Amiro et al. (2006) |
| CH-Dav | ENF | Lukas Hoertnagl | CH-Fru | GRA | Zeeman et al. (2010) | CH-Oe1 | GRA | Christof Ammann |
| CH-Oe2 | CRO | Christof Ammann | CN-Cha | MF | Shijie Han | CN-Dan | GRA | Shi Peili |
| CN-Din | EBF | Guoyi Zhou | CN-Du2 | GRA | Chen Shiping | CN-Ha2 | WET | Yingnian Li |
| CN-HaM | GRA | Kato et al. (2006) | CN-Qia | ENF | Huimin Wang | CZ-BK1 | ENF | Marian Pavelka |
| DE-Geb | CRO | Antje Moffat | DE-Gri | GRA | Christian Bernhofer | DE-Hai | DBF | Knohl et al. (2003) |
| DE-Kli | CRO | Christian Bernhofer | DE-Tha | ENF | Christian Bernhofer | DE-Lnf | DBF | Alexander Knohl |
| DK-Sor | DBF | Andreas Ibrom | ES-Lju | CSH | Penelope Serrano | FI-Hyy | ENF | Timo Vesala |
| FR-Fon | DBF | Bazot et al. (2013) | FR-Gri | CRO | Pierre Cellier | FR-LBr | CRO | Denis Loustau |
| FR-Pu | MF | Jean-Marc Ourcival | IT-Col | DBF | Giorgio Matteucci | IT-Lav | ENF | Damiano Gianelle |
| IT-MBo | GRA | Damiano Gianelle | IT-PT1 | DBF | Günther Seufert | IT-Ren | ENF | Stefano Minerbi |
| IT-Ro1 | CRO | Nicola Arriga | IT-Ro2 | DBF | Nicola Arriga | JP-SMF | CRO | Ayumi Kotani |
| MY-PSO | EBF | Yoshiko Kosugi | NL-Loo | ENF | Eddy Moors | RU-CHE | OSH | Corradi et al. (2005) |
| RU-Fyo | ENF | Milyukova et al. (2002) | RU-Ha1 | GRA | Dario Papale | US-Wi9 | MF | Jiquan Chen |
| US-ARM | CRO | Fischer et al. (2007) | US-ARb | GRA | Margaret Torn | US-ARc | GRA | Margaret Torn |
| US-Blo | ENF | Goldstein et al. (2000) | US-Cop | GRA | David Bowling | US-IB2 | CRO | Roser Mantamala |
| US-Goo | GRA | Tilden Meyers | US-Ha1 | DBF | Goulden et al. (1996) | US-Los | MF | Ankur Desai |
| US-Ivo | WET | McEwing et al. (2015) | US-MMS | DBF | Schmid et al. (2000) | US-Me2 | ENF | Campbell and Law (2005) |
| US-Me3 | ENF | Bond-Lamberty et al. (2004) | US-Ne1 | CRO | Simbahan et al. (2006) | US-Ne2 | CRO | Amos et al. (2005) |
| US-Ne3 | CRO | Verma et al. (2005) | US-Oho | DBF | Noormets et al. (2008) | US-PFa | MF | Richardson et al. (2006) |
| US-SRM | WSA | Scott et al. (2009) | US-Syv | MF | Ankur Desai | US-Ton | WSA | Chen et al. (2007) |
| US-Var | GRA | Ma et al. (2007) | US-WCr | DBF | Cook et al. (2004) | US-Wi3 | DBF | Jiquan Chen |
| US-Wi4 | MF | Jiquan Chen | | | | | | |

## 3.2 Tower data

The FLUXNET 2015 synthesis data set (http://fluxnet.fluxdata.org/) is used to obtain point-based measurements of evaporation (referred to as tower ET), and it is processed as in Martens et al. (2016) to retain only high-quality data appropriate to evaluate the evaporation estimates. Starting from the original time resolution (generally 30 minutes or 1 hour), the processing involves: (1) masking measurements using the originally provided quality flags; (2) masking measurements for rainy intervals, 280 only leaving observations if both the global precipitation product and the local measurements (if available) do not indicate precipitation (as eddy-covariance measurements are less reliable during precipitation events), and; (3) aggregating to daily values if more than 75 percent of remaining sub-hourly data exists for a given day. This quality-check yielded 97 stations. This sample was further reduced to 84 by visually inspecting aerial pictures of the tower surroundings and removing stations 285 close to water bodies, or not representative of the overall land cover within the $0.25^o$ cells of the gridded ET estimates. The geographical locations of the 84 stations, and their location in an air temperature and precipitation space, are plotted in Fig. 1, with the station names, land covers (based on

the International Geosphere-Biosphere Programme (IGBP) classification), and reference or Principal Investigator listed in Table 1. Note that nearly all stations are in Europe and US, with only two stations located in the Southern Hemisphere.

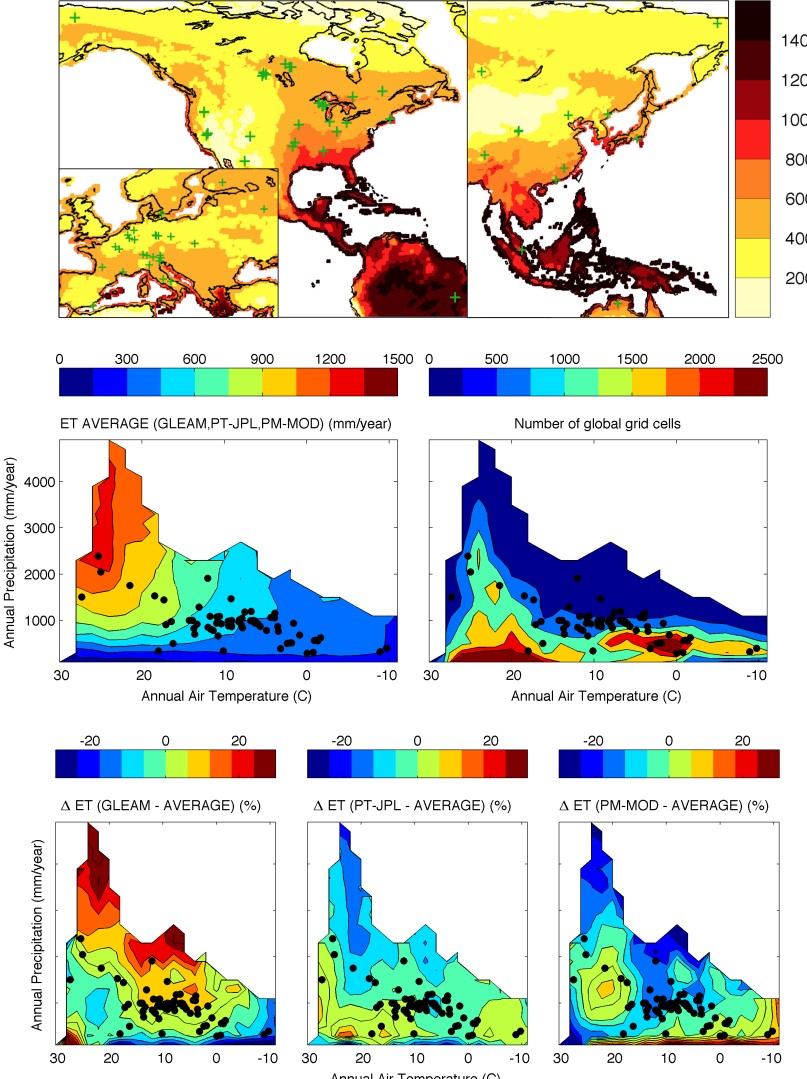

**Figure 1.** Distribution of tower sites used in the study. Top: geographical location (green crosses) on a map of the multi-annual simple average of the three ET products (GLEAM, PT-JPL, and PM-MOD). Middle: distribution of the averaged multi-annual ET (left), and the number of global grid cells (right), as function of the annual air temperature and precipitation, together with the location of the tower sites in this space (black dots). Bottom: the relative GLEAM (left), PT-JPL (middle), and PM-MOD (right) ET differences normalized by the multi-annual simple average of the three ET products.

Eddy-covariance measurements are subject to errors, both random and systematic, and any merging technique using them as reference is likely to be impacted by those errors. Systematic errors

can arise from instrumental calibration and unmet assumptions about the meteorological conditions, while random errors are typically related to turbulence sampling errors, the assumptions of a constant footprint area, and instrumental limitations (Moncrieff et al., 1996). Estimating these errors is far from simple, and typically requires dedicated experiments (Nordbo et al., 2012; Post et al., 2015; Wang et al., 2015). As such, reporting them is not a widespread practice and error statistics for the individual sites are not commonly available.

The propagation of systematic errors typically results in the lack of energy balance closure observed at many eddy-covariance sites (Wilson et al., 2002; Foken, 2008). Methods to correct the energy unbalance exist, with the Bowen ratio approach (Twine et al., 2000) and the energy balance residual approach (Amiro, 2009) being the most frequently adopted. Corrected fluxes are typically preferred over the original uncorrected observations, but these corrections implies the need for surface radiation and soil heat flux measurements, which are not routinely measured at all stations. At the sites where they are available, the FLUXNET 2015 data set offers a test product containing a corrected version of the heat fluxes based on the Bowen ratio approach, i.e. assuming that the measured Bowen ratio is correct. For the 84 stations selected here, 26 do not have Bowen Ratio Corrected (BRC) fluxes. For the remaining 58 stations, the relative mean difference between the original and BRC latent heat fluxes averaged over all stations is 6.1%, with a maximum value of 16.5%. If the correlation coefficient between original and BRC fluxes is calculated at each station and then averaged over all stations, we obtain 0.96, showing that the original and BRC ET correlate well in time. Also, if the weights of Equation 2 are calculated with the original and BRC fluxes, they display a 0.91 average correlation over all stations and models, with an average RMSD of 0.035. These numbers do not suggest strong differences between both, thus the original (uncorrected) fluxes for all stations are retained for our analyses in order to maximize the number of sites.

Moreover, not all stations cover completely the 2002-2007 period, with 6, 14, 24, 9, and 31 stations reporting 2, 3, 4, 5, and 6 years of data withing the period, respectively. At stations where interannual variability is large the weights may not be representative of the overall climate conditions at the tower if only a relatively short number of years exist. Limiting the study to stations with a relatively large number of years could minimize this drawback, but it would severely reduce the number of towers, so this filtering has not been applied. For instance, if we only derive weights for towers with at least 4 years of data, half of the towers would have been removed. Notice also that due to the masking of the tower data the 61 consecutive daily estimates required to estimate our temporally-varying weights (see Section 2.3) are generally not all available. Therefore, in the case of the tower data we set a minimum threshold of 15 daily values within the 61-day running window for the error to be estimated. Most stations have weights for nearly all days, but in a few stations there are recurrent gaps. A clear example is the tropical BR-Sa3 station, where the frequent rainy episodes complicate the derivation of the weights.

### 3.3 Ancillary data

Because the substantial mismatch between the size of the model grid cells and the tower footprint is likely to result in representativeness errors, ancillary data sets are required to characterize the spatial homogeneity of the grid cells where the stations are located. Two data sets are considered: the MODIS Land Cover Type product MCD12Q1 at a native resolution of 500 meters, and the Terra MODIS Vegetation Continuous Fields product MOD44B, available at a spatial resolution of 250

meters. A homogeneity index ($I_h$) is constructed as:

$$I_h = \frac{1}{2} Fgt_{IGBP} + \frac{1}{2}(1 - \mid Fg_{bare} - Ft_{bare} \mid - \mid Fg_{herb} - Ft_{herb} \mid - \mid Fg_{forest} - Ft_{forest} \mid) \quad (8)$$

where $Fgt_{IGBP}$ is the fraction of MCD12Q1 500 meter cells included in the 25 km model grid cell containing the tower and having the same IGBP land cover than the model cell, $Ft_{bare}$, $Ft_{herb}$ and $Ft_{forest}$ are, respectively, the bare, herbaceous, and forest fractions of the MOD44B 250 meter cell

containing the tower, and $Fg_{bare}$, $Fg_{herb}$ and $Fg_{forest}$ are the same fractions but calculated for the entire 25 km model grid cell where the tower is situated. The first term is the mismatch between the land cover at the tower and at the grid cell level, and the remaining terms are the net mismatch in land cover types across the two resolutions. $Ih$ takes values in the range [0,1], the larger the value the more representative the grid cell is of the landscape of the tower footprint. Finally, to evaluate the

merged products, we use river run-off from a compilation of monthly data using different sources, as described in Beck et al. (2015) and annual precipitation estimates from WorldClim Fick and Hijmans (2017) and MSWEP (Beck et al., 2017).

### 4 Inter-product comparison

The multi-annual GLEAM, PT-JPL, and PM-MOD total ET, together with their absolute and relative

differences, are shown in Fig. 2. Differences of the same order can be observed when other products are inter-compared (Jimenez et al., 2011). Given the use of common meteorological forcing (see Section 3.1), the observed differences are mainly introduced by the different approaches to model ET. The disagreement also extends to the the models partitioning of ET into its different components, as shown in Miralles et al. (2016) and (Talsma et al., 2018). We recall here that, as discussed in

Section 2.3, only the sum of the soil evaporation and transpiration is compared against tower fluxes.

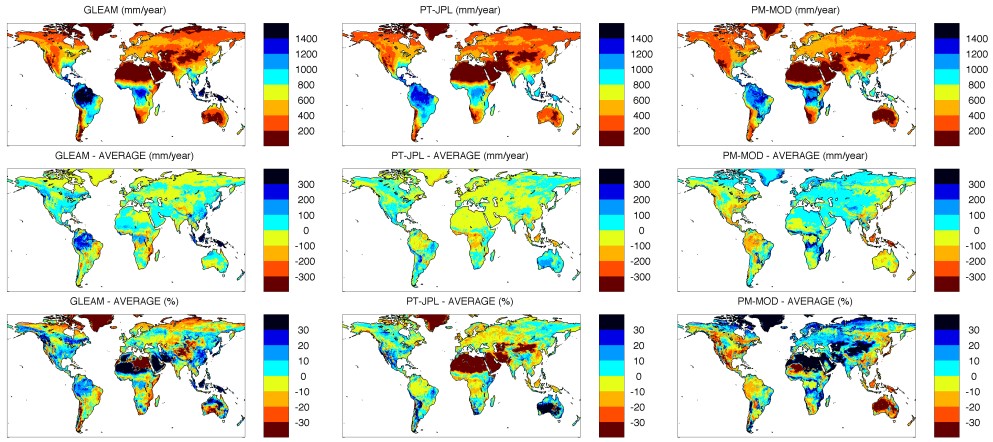

**Figure 2.** Summary of GLEAM, PT-JPL, and PM-MOD annual ET differences. Top: The GLEAM (left), PT-JPL (middle), and PM-MOD (right) total annual ET in mm/year. Middle: differences between each product and the simple inter-product mean, in mm/year. Bottom: same differences, but normalized by the inter-product mean ET, and expressed as a percentage.

Next, the ET estimates of GLEAM, PT-JPL, and PM-MOD are evaluated at the available tower sites. If we look at the towers spatial distribution in Fig. 1, we can see that there are mostly located in temperate regions. The tropical rain forest and savannas, where the relative ET differences seem larger, are less represented in the selected tower data. Therefore, some regions that would have

been relevant to characterize the model ET differences are missing in the evaluation with tower data. Seasonal distributions of ET for three vegetation classes are presented in Fig. 3. The first one includes forest stations (forest), the second one shrublands and savannas (shrub/savanna), and the third one croplands and grasslands (crop/grass). The stations are not evenly distributed within the three groups, with the forest (50 stations) being more represented than the shrubs/savanna and crops/grass (10 and

24, respectively), indicating that summary statistics could be more robust in the case of forests. The surface available energy (Ae) is also plotted. For the models, Ae is the difference between the surface net radiation and the modelled ground flux. For the towers, as the surface net radiation and/or ground flux are not measured at all towers, Ae is given by the sum of the sensible and latent heat fluxes. Clear differences between GLEAM, PT-JPL, PM-MOD and the tower probability distributions are visible.

Overall GLEAM and PT-JPL agree better with each other than with PM-MOD, which may be related to the common modelling framework of Priestley-Taylor for GLEAM and PT-JPL, compared with the more different Penman–Monteith approach of PM-MOD.

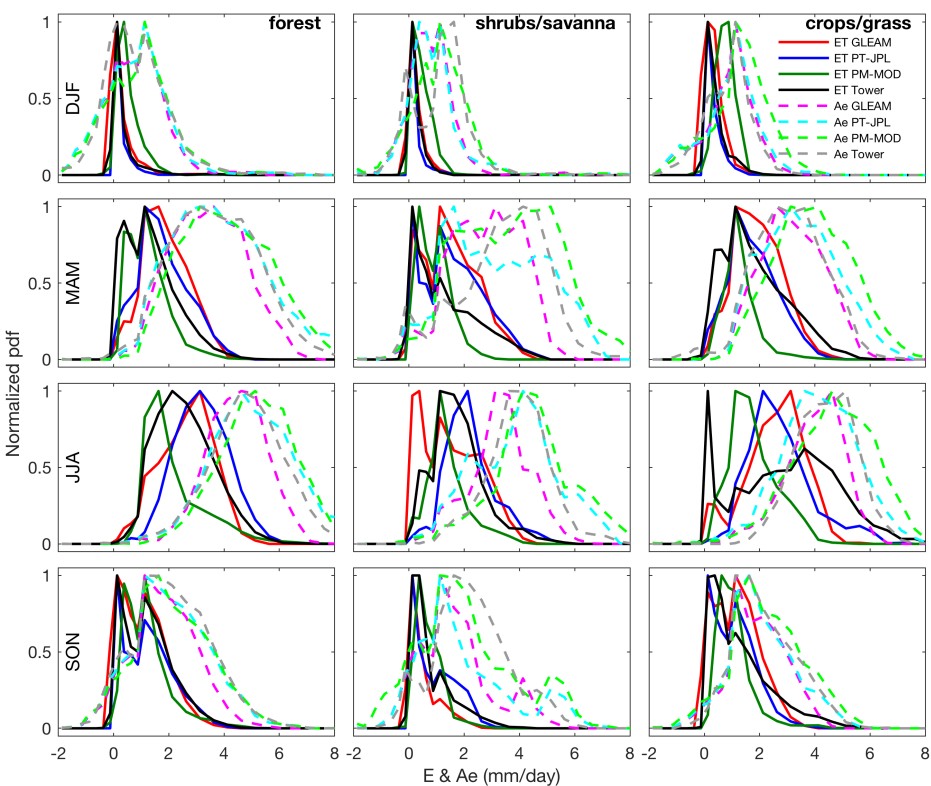

**Figure 3.** Normalized histograms of ET and available energy (Ae) from GLEAM, PT-JPL, PM-MOD, and the tower observations. The histograms are calculated with the ET values at the tower locations separated first by season and land cover.

An example of good agreement is the forest group in autumn, with the distributions of both ET and Ae being quite similar for the observed and modelled variables. The crops/grass group in summer also shows reasonable agreement between the GLEAM and PT-JPL ET distributions, but larger differences with PM-MOD and the tower ET. In that case, the tower ET shows a clear bimodal distribution, which cannot be replicated any of the models. This may be due to agricultural management practices being poorly captured by the models (e.g., irrigation), but may also reflect the large heterogeneity of croplands and their (a priori) low representativeness of the larger pixel scale. For the shrubs/savanna group during summer, the four ET distributions are quite different, with the Ae distributions also showing differences. For these cases it is difficult to identify whether tower and model ET differences are due to biases in the surface radiation, or discrepancies in the ET formulations.

## 5 Local merging

### 5.1 Local weights

A summary of daily weight statistics over all the sites belonging to a given land cover group is given in Fig. 4. These weights have been derived based on the differences between the ET product anomalies and the tower ET anomalies as explained in Section 2.2.1. As expected, the simple average product (SA-merger) equally weights all products with a value of 1/3 and is added here as reference. Notice that the weights can take negative values, although the sum of the weights is still one. This

happens when the full error covariance matrix has large off-diagonal values reflecting the correlation between the different product errors (e.g. Jones et al., 2008; Hobeichi et al., 2018). This correlation is expected given that the products share some common inputs and model formulations, and it is specially noticeable for GLEAM and PT-JPL. On average, GLEAM has the largest weights and contributes more to the weighted anomalies, but the relative weight of each model is not uniform

per season or land cover. For instance, for the forest class PT-JPL is more weighted than GLEAM in winter, while the reverse is true in autumn.

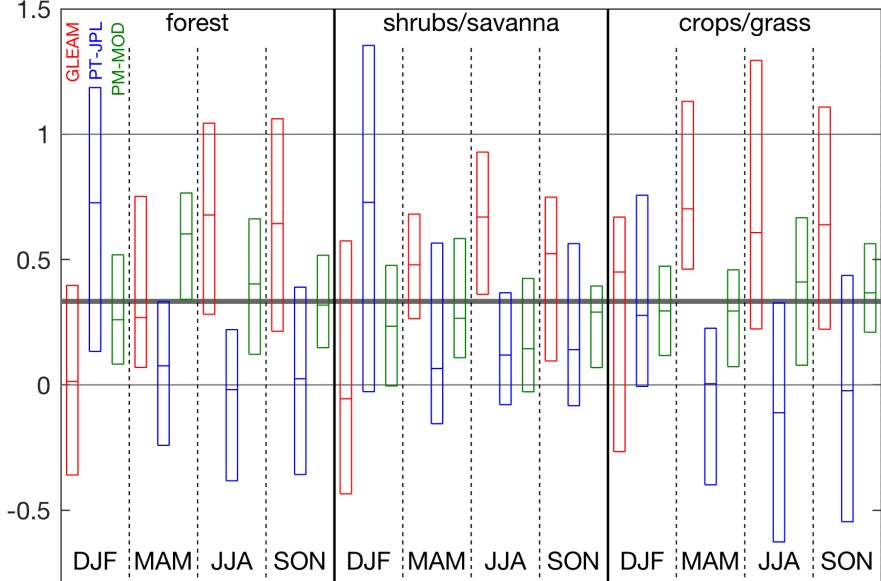

**Figure 4.** Box plots of the GLEAM (red), PT-JPL (blue), and PM-MOD (green) seasonal weights for the three land cover groups. The central mark of the box plots is the median of the group population, the box edges are the 25th ($Q1$) and 75th ($Q3$) percentiles.

An example of the temporal variability of the weights at three towers is given in Fig. 5. At the FR-Pue site, a Mediterranean forest located in France (Rambal et al., 2004), GLEAM starts to be

clearly more weighted for the second part of the year. The correlation between the GLEAM and PT-JPL anomalies is visible in the anti-correlation displayed by the weights. At the US-SRM site, a semi-arid grassland site in southwest of US (Scott et al., 2009), PM-MOD is typically more weighted than GLEAM and PT-JPL in spring, and all weights depart less from the 0-1 range, suggesting more independent errors at this particular station. The last site, the US-Ne1 cropland station situated in North America (Verma et al., 2005), is an example of closer weights for all models for some periods of the year. This happens during the first half of the year. For the second part of the year, the weights change more, with PT-JPL being the most weighted product during some months.

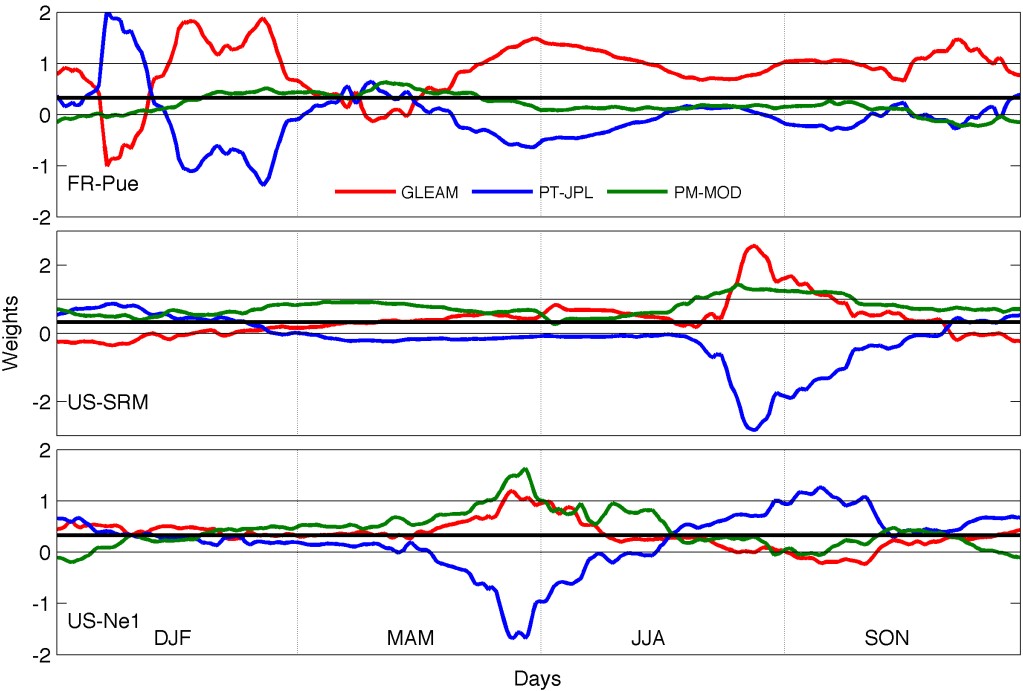

**Figure 5.** Example of GLEAM (red), PT-JPL (blue), and PM-MOD (green) weights at the FR-Pue (top, forest), US-SRM (middle, grassland), and US-Ne1 (bottom, cropland) stations. The thick black line marks the 1/3 value of the SA-merger weights; the thin black lines mark the 0-1 interval.

## 5.2   Merged products

Fig. 6 shows – for the same three towers in Fig. 5 – time series of ET from the three products, SA-merger and the weighted average (WA-merger), and the in situ measurement for 2006. At the FR-Pue site, for this specific year all products disagree with the tower ET for a large part of the year, with PT-JPL and PM-MOD having much larger absolute values overall. Differences between SA-merger and WA-merger are mainly visible in spring and summer, where GLEAM is weighted more strongly, making WA-merger follow the GLEAM estimates more closely. The US-SRM site shows a relatively large ET seasonal variability, with the ET tightly linked to the precipitation and associated

increases in soil moisture (Scott et al., 2009). GLEAM and PT-JPL capture this variability, especially the sudden increase in ET values at the beginning of summer related to the rainfall coming from the North American monsoon. For the first half of summer there are sometimes large differences between SA-merger and WA-merger, with WA-merger correlating better with the tower ET. For the second half, all products fail to replicate the ET increase measured by the tower, and WA-merger and SA-merger are closer to each other as the models anomalies cannot provide information to guide the merging. The US-Ne1 is an irrigated maize-soybean site, where the seasonal cycle of ET is expected to be more pronounced, and accompanied by higher absolute values resulting from irrigation (Verma et al., 2005). The original products have more similar values, not capturing well the ET rise associated with start of the growing season. This may have to do with irrigation not being well captured by any of the models. The closer weights shown in Figure 5 result in closer SA-merger and WA-merger ET.

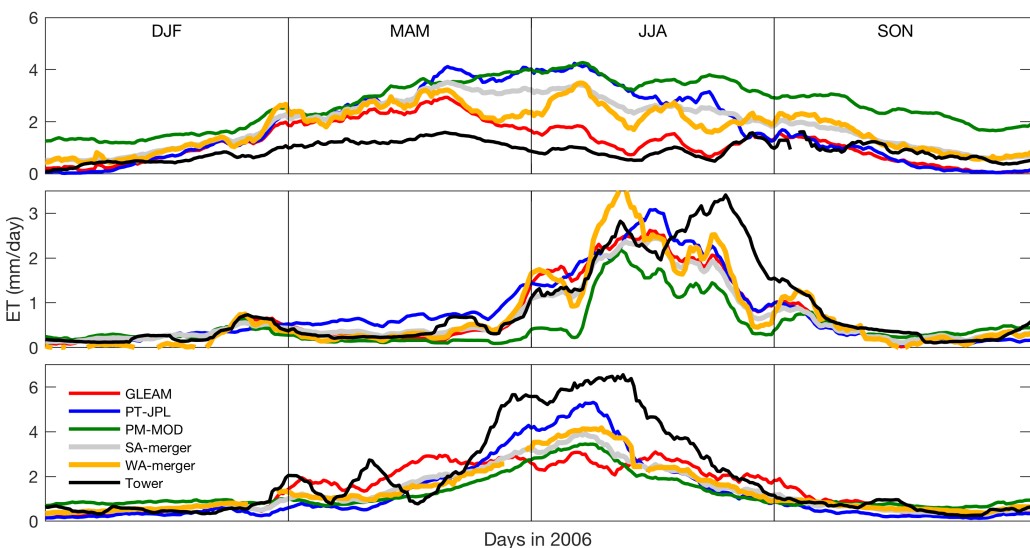

**Figure 6.** 2006 time series of the different ET products and the sites shown in Fig. 5: FR-Pue (top), US-SRM (middle), and US-Ne1 (bottom). The daily values are time smoothed using a 10-day moving averaged window to better display the more persistent temporal features.

The performance of the individual and merged products across the different stations is summarized in Fig. 7 by plotting seasonal averaged correlations and RMSDs for the three land cover classes. The statistics are presented for the ET anomalies, and for the absolute values. For both, in 10 out of the 12 cases presented (4 seasons x 3 land covers) the correlation of WA-merger is higher than for SA-merger, indicating that an appropriate characterization of the errors – and derived weights – results in better estimates of the ET. The relative increases in correlation between SA-merger and WA-merger are larger for the ET anomalies, but still occur for the ET absolute values. This highlights that when the weighted anomalies are added to the multi-product climatology, the resulting product

combination still overcomes the simple average. Note that the lowest correlations occur in winter time, reflecting the low values and low intra-seasonal variability in this period, while the largest correlations are observed in spring and autumn where vegetation greening and browning typically results in larger ET variability. Note also that correlations are not significant for some stations and periods; non-significant correlations are typically found in wintertime.

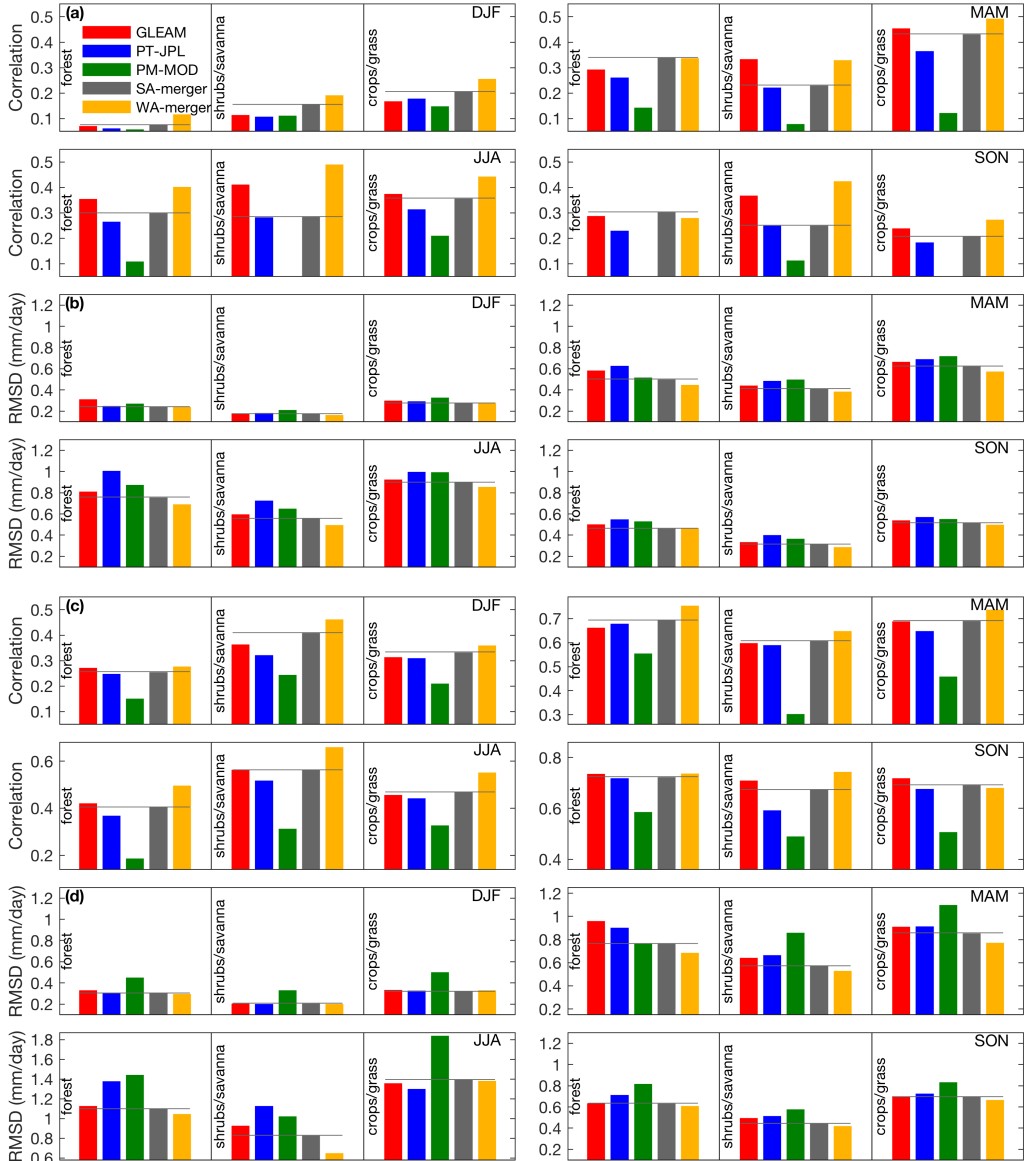

**Figure 7.** Season and land-cover averaged ET correlations and RMSD of the tower and the different products (a and b for the ET anomalies, c and d for the ET absolute values). To highlight differences with SA-merger, a grey line has been added to its bar. Note that the axes are not identical, but they cover similar ranges (0.5 for the correlation, 1.2 mm/day for the RMSD).

Concerning the RMSDs, they are slightly lower for WA-merger for all seasons except for winter months. As SA-merger and WA-merger share their climatology (see Section 2.2.1), large differences between both are not expected. This means that the biases between the merged products and the tower ET are preserved for both mergers, indicating that most of the differences in RMSD is coming from changes that are also reflected in the correlations.

# 6  Global merging

## 6.1  Global weights

The local weights at the 84 stations have been extrapolated by the NN as described in Section 2.2.2. The seasonal averages of the weights are presented in Fig. 8. Overall, the spatial patterns of the extrapolated weights for each product do not change substantially across the seasons. Some exceptions

are Europe and Northern Asia for GLEAM and PT-JPL. The PM-MOD weights are mostly positive, apart from forested areas in the tropics and some dry areas in Asia and Australia, and are more confined than GLEAM and PT-JPL to the 0-1 interval, indicating smaller error correlation with the other products. For GLEAM and PT-JPL, the weights are a mixture of positive and negative values, and a clear anti-correlation of the weights is visible, i.e., positive GLEAM weights correspond to negative

PT-JPL weights, and vice versa, similar to the pattern observed in the local weights for some periods.

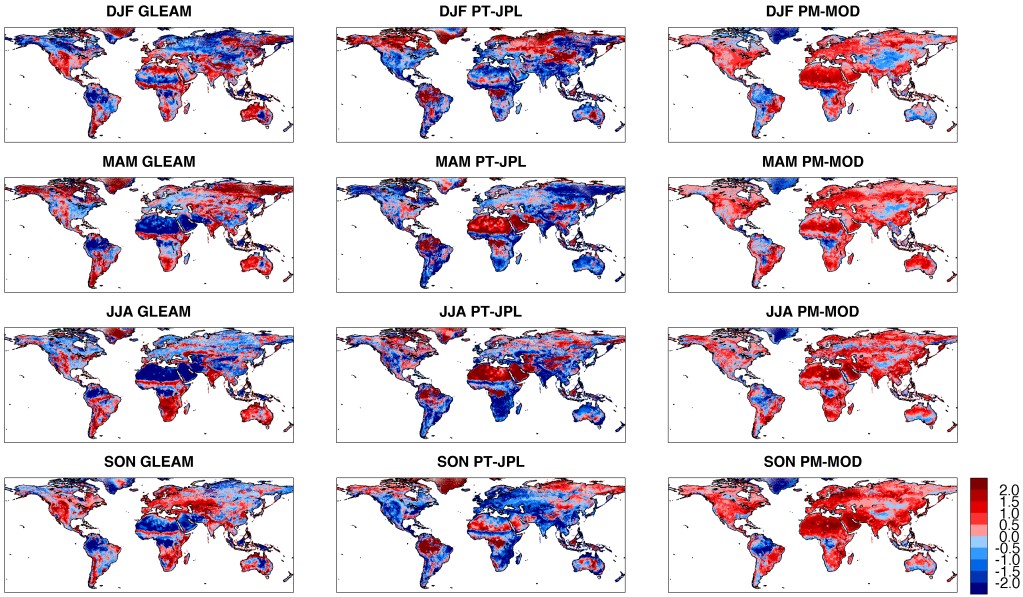

**Figure 8.** Seasonally averaged global weights for GLEAM (left), PT-JPL (middle), and PM-MOD (right). Red (blue) colours indicate positive (negative) weights.

## 6.2 Merged products

The seasonally averaged ET differences between WA-merger and SA-merger, normalized by the seasonal SA-merger, are plotted in Fig. 9. The large differences in (semi-)arid areas or the northern latitudes in winter are related to the very low ET absolute values. For the remaining land, most of the relative differences are within the ±25% range. Overall, there are more negative than positive differences, indicating that the WA-merger results in smaller absolute values. Giving that SA-merger and WA-merger have a common climatology, this suggests that the weighting results in an overall reduction of the anomalies at many regions.

Some geographical structures and seasonal changes are visible in some regions. For instance, in the sub-Saharan transition zone the differences are positive in the first half of the year (WA-merger > SA-merger), but negative in the second half. Over India the differences are positive in autumn and winter, but negative in spring and summer. In contrast, some regions do not display large seasonal changes. For instance, in most of Europe WA-merger is smaller than SA-merger over all seasons.

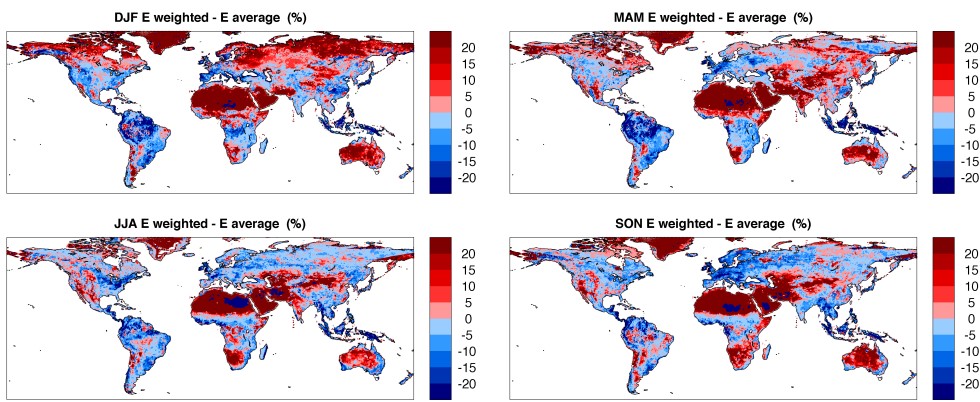

**Figure 9.** Seasonally averaged normalized ET differences between SA-merger and WA-merger, expressed as a percentage of the seasonally averaged SA-merger ET. Red (blue) colours indicate positive (negative) differences.

## 7 Discussion

### 7.1 Tower representativeness

Our inverse error variance weighting is based on the differences between the model and tower ET anomalies. However, it is expected that part of the difference between in situ measurements of ET and model estimates respond to the mismatch in spatial resolution (tower footprint versus model cell). The RMSD of SA-merger against the towers ET, normalized by the mean annual tower ET, is

displayed in Fig. 10 for all the available stations, together with the station $I_h$ described in Section 2.3. The towers are sorted from maximum to minimum $I_h$, i.e., starting by the towers better representing the grid cells where they fall. Nonetheless, low and high normalized RMSDs can occur at stations with comparable $I_h$, indicating that spatial heterogeneity is only one of the contributing factors to the ET differences. In fact, if the RMSD is linearly regressed on the $I_h$, the slope of the fit is close to zero, as shown in Fig. 10. Also for the separate products (GLEAM, PT-JPL, and PM-MOD) and WA-merger, no significant correlation between their RMSD against in situ measurements and Ih was found (results not shown). This indicates that for the calculated Ih, and the selected sample of ET products and stations, the error related to the inconsistencies between the tower footprint, and the model pixels does not dominate the total error budget.

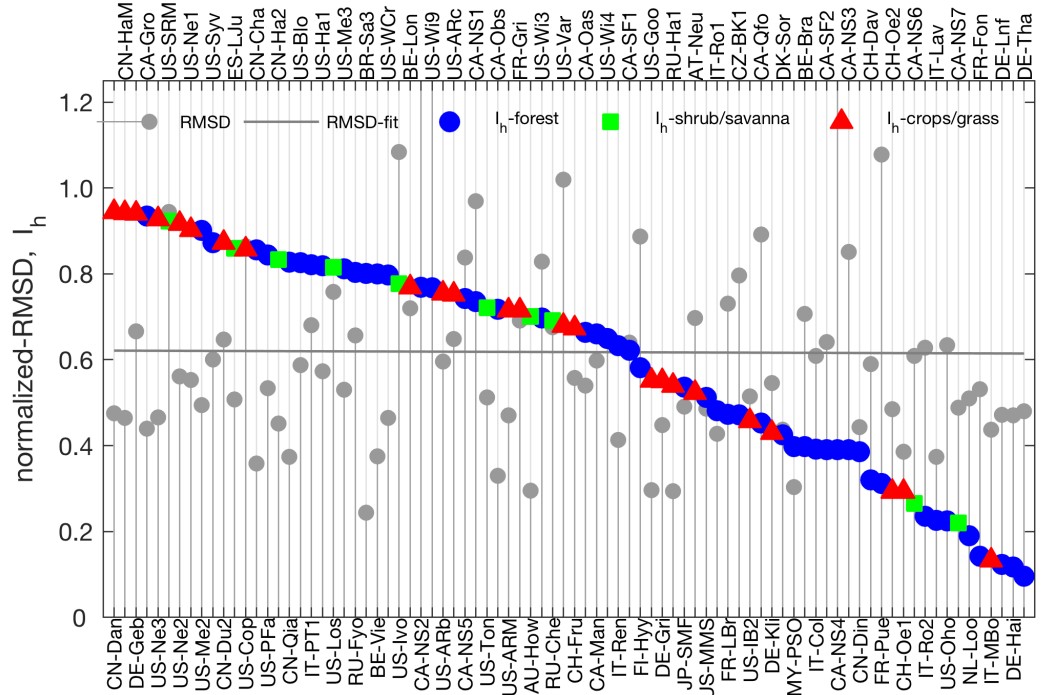

**Figure 10.** Homogeneity index ($I_h$) and RMSD of SA-merger and the towers ET. The $I_h$ is plotted as closed circles in blue for forest stations, green for shrubs/savanna, and red for crops/grass, while the RMSD, normalized by the mean annual tower ET, is plotted in grey. A linear fit to the normalized RMSD is given by the grey line. The towers are sorted from maximum to minimum $I_h$, with the tower names given at the bottom and top of the figure.

## 7.2 Inverse error variance weighting

The objective on an inverse error-variance weighting is to find the estimate that minimizes the variance of the random error (Rodgers, 2000). As such, the merging only results in the optimal weights if applied over an ensemble of unbiased estimates. Strictly speaking, this requires removing the bias

between the model ensemble and in the situ observations prior to the merging, which is not the case here (see Equations 1 to 3). The objective here was to correct the product anomalies towards the tower anomalies, but not to correct the original estimates toward the tower in absolute terms. On the one hand the tower observations have their own systematic errors, as discussed in Section 3.2. On the other hand, debiasing toward the tower ET would require a global correction of the gridded products towards a global tower climatology. If the ultimate objective is to reproduce the tower fluxes, other approaches like regressing the tower ET on either the ET products (Yao et al., 2017) or the ET explanatory drivers (Jung et al., 2011) may appear more straightforward and be possibly more appropriate.

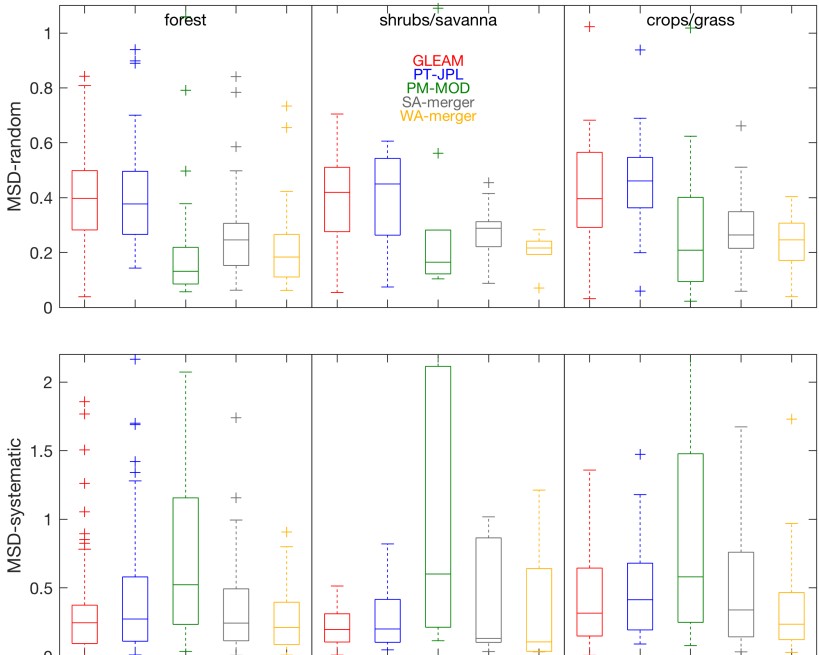

**Figure 11.** For the three land cover groups the random (top) and systematic (bottom) MSD between the tower ET and GLEAM (red), PT-JPL (blue), PM-MOD (green), SA-merger (grey) and WA-merger (yellow). The central mark of the box plots is the median of the group population, the box edges are the 25th ($Q1$) and 75th ($Q3$) percentiles, the whiskers extend to $Q3 + 1.5(Q3 - Q1)$ and $Q1 - 1.5(Q3 - Q1)$, and values outside the whisker are plotted individually.

Nevertheless, even if optimality in the sense of minimizing the error variance of the WA-merger cannot be assured, weighting the anomalies should result in a decrease of the random error. This is shown in Fig. 11, where box plots of the random ($\mathrm{MSD}_r$) and systematic ($\mathrm{MSD}_s$) components of the difference between the products and the tower observations are displayed (see Equations 6 and 7). From the original products, GLEAM and PT-JPL have comparative error components, while

PM-MOD is more distinctive, having smaller $MSD_r$ and larger $MSD_s$. The latter likely relates to the tendency of the PM-MOD to underestimate ET and its variance (Michel et al., 2016; Miralles et al., 2016). Comparing WA-merger to SA-merger, the reduction of the $MSD_r$ for WA-merger is indicative of the merging being effective in this regard. There is also a slight reduction in the $MSD_s$, with WA-merger having the smallest median error of all products.

### 7.3 Weights extrapolation

The number of stations used in this merging exercise is certainly limited in terms of covering different biomes and climatic conditions. Hence, the ability to represent the full distribution of ET across time, space, and biomes is questionable. This is verified here by out-sampling the NN training data set in two different ways. In the first test all stations are included in the tower data set, i.e., the standard configuration used to produce the global WA-merger. Before training the NN, 15% of the days at each station are randomly masked from the training data set, and the prediction statistics are derived over this independent subset. In the second test, the station where the prediction will be checked is entirely removed from the training data set, i.e, the weights for that station are derived using a NN that did not include that station in the training phase (i.e. leave-one out cross validation).

A box plot summarizing the correlation and RMSD between the station weights and the weights predicted by the NN for these two tests is presented in Fig. 12. The results clearly show that the correlation and RMSDs between the predicted and the original weights at the stations degrades notably when stations are fully removed from the training data set. This implies that the global extrapolation of the weights will be quite uncertain for conditions not sampled in the available tower data set. For some stations, the out-sampling from the training data set does not have a large effect, because the mapping between the predictors and ET can still be approximated from the relationship presented by other stations. This is for instance the case for the Canadian forest stations CA-NS1-7 (results not shown). However, for other stations, the statistics are good when predicted with the standard data set, but poor with the one-station-removed data set, indicating that the particular conditions of those stations are not well represented in the out-sampled data set. This happens for stations such as US-Wi4 (forest with a snowy winter and warm humid summer) and CN-Dan (grasslands with a polar tundra climate). Finally, there are also stations where statistics are rather poor in both tests, indicating that a link between the model inputs and the related output error could not be established. This is the case for stations such as IT-Col (deciduous broadleaf forest with temperate climate) or MY-Pso (tropical forest). This guaranties that the extrapolation of weights to areas with similar conditions will be very uncertain, even if those conditions were represented in the tower data set.

An additional test to check the representativeness of the tower data set is conducted by globally extrapolating the weights with each of the previous 84 NNs trained without one station, and then checking the variability of the predicted weights. For the conditions well represented in the training data set, it is expected that removing one station will only result in slight changes in the extrapolated

weights. However, for regions that are poorly represented, a slightly different data set is likely to result in substantially different weights. This is illustrated in Fig. 13, where a weight variability index is displayed. The index is calculated by: (1) estimating for each global cell the annual standard deviation of the GLEAM, PT-JPL, and PM-MOD weights, normalized by the sum of the absolute annual model weights, and; (2) averaging this standard deviation over the three models. To facilitate its display in Fig. 13, it has been scaled to span the range 0-1. Overall the variability is larger in the Southern Hemisphere than in the Northern Hemisphere, which is expected given that all stations but two are situated in the Northern Hemisphere. The smallest variability in the weights coincides with the regions where the database is more representative, namely US, Central Europe, and some parts of Asia, suggesting a bias in the tower data set linked to the specific location of the towers selected. The variability in tropical regions, where only 3 stations are part of the database, is in general larger than for the previous regions. The largest variability occurs over the very dry regions, a regime poorly represented in the tower data set as shown in Fig. 1. While a poor extrapolation of weights is not critical over very dry regions, given their low ET values, uncertain weights over the very humid regions is more of a concern due to their typically large ET values and their significance for the global mean ET.

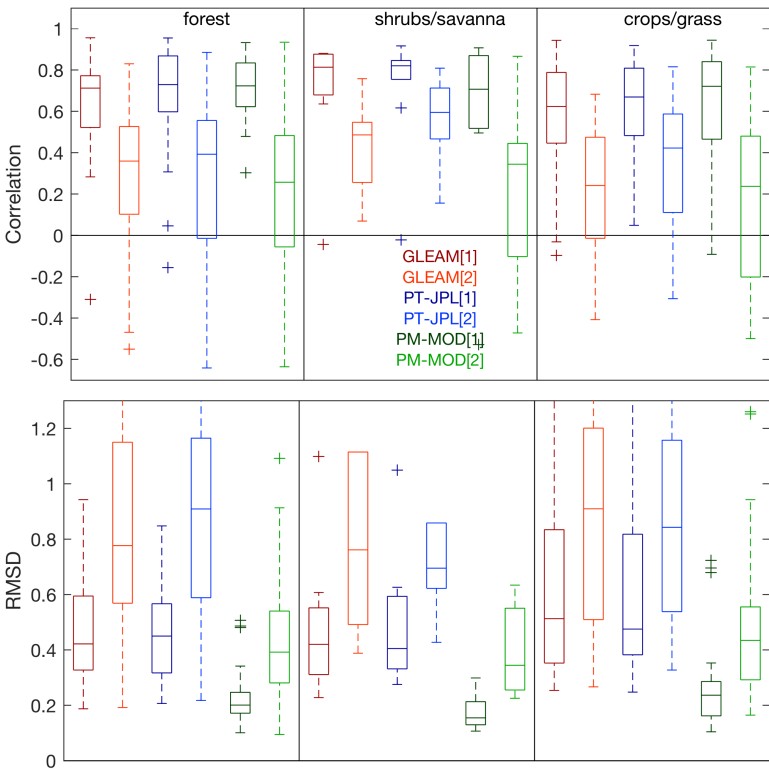

**Figure 12.** Box plot showing for the three land cover groups the correlation (top) and RMSD (bottom) between the station local weights and the weights predicted by the NN for the two tests presented in Section 7.3 (first test labelled as [1] in legend, dark colours, second test as [2], light colours). See the text for details.

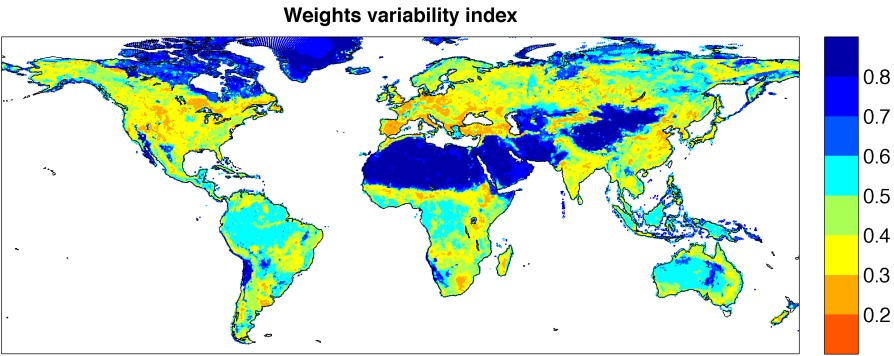

**Figure 13.** Relative annual variability of the global weights extrapolated by 84 different NNs. Smaller (larger) values indicate lower (higher) variability. See the text for details.

## 7.4 Merged products evaluation

The evaluation of ET products is typically conducted by comparing the estimates to point scale tower fluxes. Alternatively, water balance calculations at larger spatial scales – such as catchment scales - where ET is estimated as the residual of precipitation (P) and river run-off (Q) are often used as well. As the towers are used to derive the merge products, the alternative for an independent assessment of the merged products is to conduct such catchments mass balance analyses (e.g. Vinukollu et al., 2011; Miralles et al., 2016). This assessment appears a good means to evaluate the long-term mean ET estimates. Nonetheless, as WA-merger and SA-merger share a common mean state, large performance differences are not expected. Note that to retain the independence, the precipitation used in the water balance calculation should not be the one used as forcing in the ET estimates. Here this is an issue for GLEAM and the merged products (as they include GLEAM), but not for PT-JPL and PM-MOD as they do not use precipitation data as input. As such, WordlClim precipitation data is also used in addition to MSWEP in these comparisons (GLEAM was forced with MSWEP, see Section 3.1).

The mass balance of a catchment implies that the space and time integration of P-Q should equal the ET integrated over the same space and time, if one assumes that the the changes in soil water storage within the catchment are small compared with the cumulative volume of ET, P, and Q. The longer the period, the more valid this assumption becomes. Here, the mean 2002-2007 ET estimates from GLEAM, PT-JPL, PM-MOD, and the merged products are calculated per catchment. The basin P-Q estimate is then calculated using the Q and P data described in Section3.3. We only select catchments for which the P-Q data record is available for a minimum of 3 years in the 2002-2007 period, to assure some common period between ET and P-Q. In addition, to reduce noise in the basin-integrated ET estimates, only basins with a catchment area containing at least 3x3 cells of the

25 km resolution gridded estimates are included in the comparison. This results in 685 basins, 75 % of them situated in the Northern Hemisphere (i.e showing a similar geographical bias as the tower data set). Catchments are further divided into three groups of 243, 295, and 147 basins based on the aridity index (AI, basin potential ET over the basin P) taking values in the intervals AI<1, 1<AI<2, and AI>2.

Scatter plots showing the correspondence between P-Q and ET are given in Fig. 14. Linear fits for the three AI classes are plotted, and the correlation, RMSD, and bias (ET minus P-Q) given in the plot. Overall, the statistics of the the water balance comparison using MSWEP or WordlClim as P are close, suggesting that the dependence on MSWEP is not a determining factor in the agreement. From the original products, PM-MOD shows the worst agreement with P-Q. GLEAM agrees better than PT-JPL for the wettest and specially for the driest basins. For the latter, GLEAM shows correlations of 0.93 (based on MSWEP) and 0.88 (based on World-Clim), compared to the respective 0.74 and 0.69 for PT-JPL. However, PT-JPL agrees slightly better than GLEAM for the 1<AI<2, although both show similar correlations. However, PT-JPL agrees slightly better than GLEAM for the 1<AI<2, although both show similar correlations. The SA-merger shows close statistics to GLEAM and PT-JPL, so adding the PM-MOD product neither improves nor degrades the skill to close the catchment water budget. Regarding a comparison between WA-merger and SA-merger, their statistics are very close. Correlations are comparable, and only in terms of RMSD WA-merger ET agrees slightly better with P-Q for the wettest basins (AI>2), with 89/83 (MSWEP/WordlClim) mm/year RMSDs and -64/-46 mm/year biases for the WA-merge product, and 115/107 mm/year RMSDs and -97/-80 mm/year biases for the SA-product. Notice also that for the wettest basins these WA-merger performs better than any individual product.

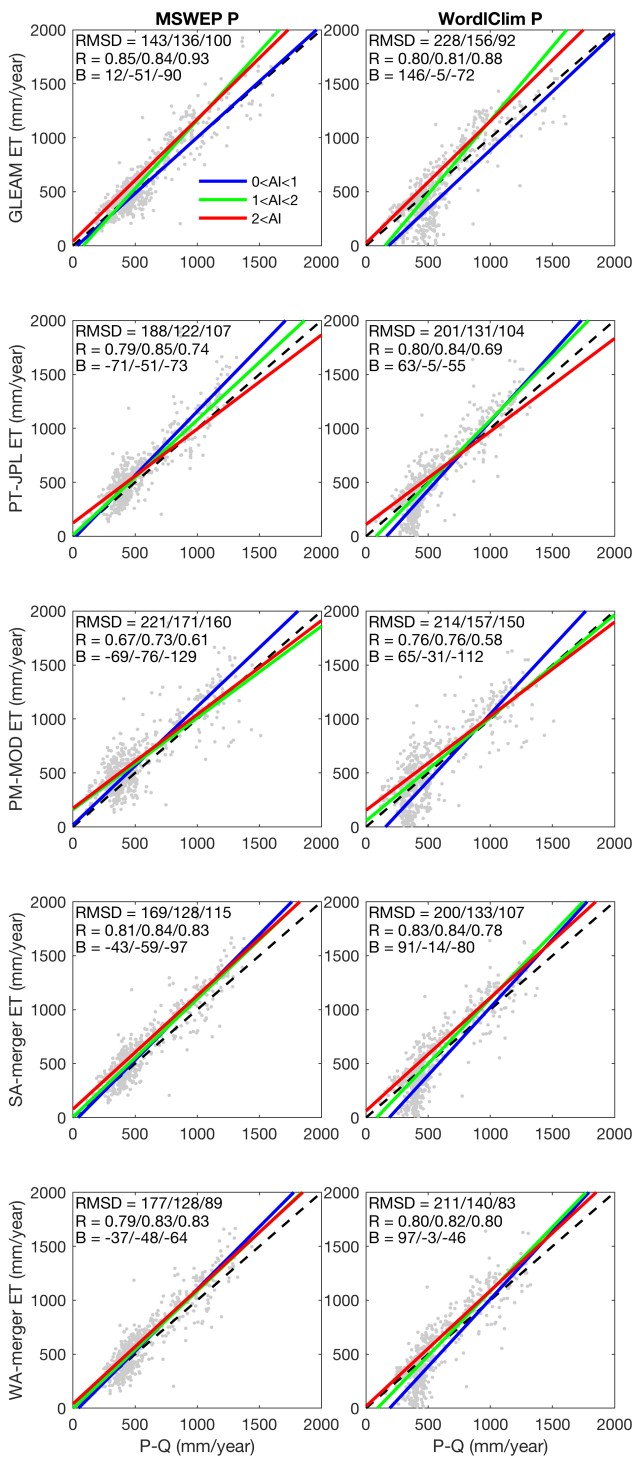

**Figure 14.** Scatter plots of P-Q and ET from the different products. Linear fits for three AI classes are plotted, together with the correlation, RMSD and bia s (ET - (P-Q)). From left to right, the statistics are given for AI<1 (blue line), 1<AI<2 (green), and AI>2 (red), i.e., from dry to wet basins.

## 8 Conclusions

A simple average (SA-merger) and an inverse error variance weighting (WA-merger) of the three global ET products generated during the WACMOS-ET project is presented. During the project, three ET models were forced with common daily inputs at a resolution of 25 km for the period 2002-2007: GLEAM, PT-JPL, and PM-MOD. GLEAM and PT-JPL share a Priestley-Taylor formulation to estimate potential evaporation, while PM-MOD uses a more different modeling approach of potential evaporation based on a Penman-Monteith formulation, but a very similar evaporative stress and radiation partitioning formulation to the one by PT-JPL. In WA-merger, the weights were estimated using the error-variance of the individual product anomalies, with the error defined as the difference between tower-based ET anomalies and modeled ET anomalies for non-rainy conditions. Then the final data set was reconstructed by adding the weighted anomalies to the mean seasonal climatology of the products. A similar approach was followed to generate SA-merger, but in this case giving equal weights to the anomalies of all three products. Finally, the potential to extrapolate these locally-estimated weights to the global scale based on a neural network approach has been explored. Given the described framework, the intent here is to evaluate the potential of blending these data sets to yield anomalies of ET that better represent those measured by the global network of eddy-covariance towers. We note that capturing anomalies in ET is crucial for applications such as drought monitoring or irrigation planning.

The resulting local weights showed seasonal patterns and negative values at many stations. This was to a large extent related to correlation in the errors of the anomalies of GLEAM and PT-JPL. Nonetheless, seasonal correlations between WA-merger and the tower ET are overall higher than for the individual products and SA-merger. This is mostly attributed to a successful reduction in the random error. Meanwhile, the globally extrapolated weights showed seasonal and regional variability, with these patterns resulted in seasonal differences between the global SA-merger and WA-merger of up to 25% in a large number of regions. However, the limited global coverage of the tower stations, mostly located in the Northern Hemisphere temperate regions, casted doubts on the ability of the NN prediction scheme to reliably extrapolate the locally-estimated weights. This was apparent when the extrapolation was tested over individual stations with the training data set not including the station under study, and when reproducing the global extrapolation of the weights with the training data set missing one station at a time. Both mergers were also compared with the ET inferred from water balance calculations in different catchments across the globe, and similar correlations and RMSDs were obtained, with only slightly better results for the WA-merger over wet basins.

Several limiting factors for the merging exercise are identified, some of which could be informative for other initiatives aiming to blend ET data sets. A longer study period can give access to more in situ data and extend the in situ data set to less represented regions. This would clearly help the global extrapolation of the weights. In addition, the mismatch between the spatial resolution of the towers and the products is still an issue, despite the fact that here other error sources were deemed to

be more dominant. The impact of the mismatch in spatial resolution is expected to be minimized as ET datasets move towards finer spatial resolutions. Dependency between the ET products can also have an impact on the merged products. In this study the GLEAM, PT-JPL, and PM-MOD products are derived with common data sets for their shared inputs. While this was motivated by the primary objective of WACMOS-ET of studying algorithm differences, this is can become a drawback when aiming to achieve an optimal merger. In that case a lower inter-dependency is expected to be beneficial.

Overall, our study suggests that an inverse error variance scheme combining information from tower observations and ET products has the potential to improve upon the simple mean proposed by several previous efforts (e.g. Mueller et al., 2013). However, care should be taken regarding the dependence of the products to be merged, the tower coverage, the different product errors, the spatial representativeness of the in situ measurements at the products resolution, and the nature of the errors of the ET products. Critical for the success of the merging scheme is the adequate characterization of the uncertainty of the individual products, and finding an effective method to extrapolate the weights from the tower space to the global landscape. The latter seems challenging, and given the difficulties found here, alternatives should be considered. A possibility could be triple collocation (Yilmaz et al., 2012). This technique would require two new global ET data sets independent from the products that need to be merged . This can be demanding, but work in that direction has already started (Khan et al., 2018). An added advantage of this approach will be that the tower observations could then be used as an independent evaluation set, similar to the approach carried out for some other Earth Observation products, such as the soil moisture estimates from the ESA Climate Change Initiative (Gruber et al., 2017). This can be of importance, given the very few existing data sets that can be used to presently evaluate ET estimates.

*Acknowledgements.* This study was funded by the European Space Agency (ESA) and conducted as part of the project WACMOS-ET-Ensemble (ESRIN Contract No. 4000117355/16/I-NB). D.G.Miralles acknowledges support from the European Research Council (ERC) under grant agreement number 715254 (DRY-2-DRY). J.B. Fischer contributed to this paper at the Jet Propulsion Laboratory, California Institute of Technology, under a contract with the National Aeronautics and Space Administration. California Institute of Technology. Government sponsorship acknowledged. Support to J.B Fisher was provided by NASA's SUSMAP, THP, and INCA programs, and the ECOSTRESS mission. K. Tu, from the Department of Ecosystem and Conservation Sciences, University of Montana, is acknowledged by providing guidance for the use of the vegetation products in this study. This work used eddy covariance data acquired by the FLUXNET community and in particular by the following networks: AmeriFlux (U.S. Department of Energy, Biological and Environmental Research, Terrestrial Carbon Program (DE-FG02-04ER63917 and DE-FG02-04ER63911)), AfriFlux, AsiaFlux, CarboAfrica, CarboEuropeIP, CarboItaly, CarboMont, ChinaFlux, Fluxnet-Canada (supported by CFCAS, NSERC, BIO-CAP, Environment Canada, and NRCan), GreenGrass, KoFlux, LBA, NECC, OzFlux, TCOS-Siberia, USCCC. Data and logistical support for the station US-Wrc were provided by the US Forest Service Pacific Northwest

Research Station. The FLUXNET eddy-covariance data processing and harmonization were carried out by the ICOS Ecosystem Thematic Center, the AmeriFlux Management Project, and the Fluxdata project of FLUXNET, with the support of CDIAC, and the OzFlux, ChinaFlux, and AsiaFlux offices.

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
