# Peer review of "Exploring the merging of the global land evaporation WACMOS-ET products based on local tower"

_Hydrology and Earth System Sciences, 2017_

## Referee Comment (RC1) · Anonymous Referee #1 · 13 Dec 2017

Review comments for 'Local tower-based merging of two land evaporation products' by C. Jimenez et al.

This study has addressed the difficulties in merging different ET products – in particular when the spatial resolutions are vastly different – those of coarse grid (25 km) versus in-situ observations (which are fetch dependent).

The detailed description of the methodologies and the analysis of results as well as the discussion are very helpful to help the reader understand the challenges in such undertaking. The conclusions are honestly drawn based on the results.

On the basis of the above facts, I recommend the publication at HESS after some minor revision.

Given the issues raised by the manuscript, which mentions issues like - dependence of the products to be merged, the tower coverage, errors, and spatial representativeness of their measurements at the products resolution, and the nature of the ET product errors, I would suggest to use a more general title e.g. 'On issues in local tower-based merging of land evaporation products' or something similar. The current title is specific but I am not sure that the merged product is more useful than each of them individually and the true contribution of the study is to enlist and highlight these issues to the community.

P8L24-25: I was not sure what 'a station-averaged square temporal correlation of 0.96.' – is this the coefficient of determination?

P12L15: I was not sure what 'the satellite surface meteorology' refers to.

---

## Referee Comment (RC2) · Anonymous Referee #2 · 13 Dec 2017

The authors investigate the added value of merging two land ET products based on their performance with respect to tower-based ET. This is definitively an interesting topic in particular in the light of the existing large uncertainties in ET estimations.

General comments and questions:
The study is well written and provides interesting insights in the performance of the two used ET models. These seem to perform very similar and the merge of them does not provide a significant added value. I'm wondering thus if the use of other WACMOS-ET models with more diverse performance at the tower sites could be a better test case for the proposed merging procedure (instead of having two already similarly well-performing models with not much of room for improvement). Can the authors comment, why they did not include a more diverse palette of models?

Also, how are data gaps in tower data and non-consistent temporal coverage of the towers treated? This might influence the analysis of the derived merging weights if the station sample changes over time.

Specific comments:
page 3, line 12: What about other observational data? E.g. lysimeters or catchment wide estimates could be mentioned as well here.
page 5, lines 24/25: Do you expect an impact on the merging weighs when sub-daily simulations and tower data would be considered?
page 7, line 18: Obsolete brackets between the two cited papers.
page 8, line 1: What happens if the two data sources for precipitation disagree?
page 12, line 8, "estimated over the time series of available errors": What about differences in the length of the EC time series or differences in the occurrence of data gaps between the towers? How is this taken into account in the analysis of the weights?
page 13, lines 3-5: Rephrase: "A 61-day running window was found to provide ..."
page 13, lines 3-5: Is there a minimum requirement of data availability for deriving the weight within the window?
page 13, lines 9-13: Sounds a bit confusing and not so clear (at this point of the paper at least). Try to re-formulate being a bit more specific (reasons to believe that?).
page 14, lines 14-20, "optimum product": How often is the optimum product one of the two models (i.e., weight of one)? From Fig. 5 it looks like a weight of one is never reached.
page 19, Fig. 4: The legend interferes with the figure information, please increase the y-axis range a bit.
page 19, Fig. 4 caption: Is it the optimum product you are comparing with the tower ET, not the WA-product? From page 14, I get that the optimum product is either the tower ET or one of the two models. Assuming that it's often the tower ET (judging from Fig. 5 where weights never reach one), shouldn't the RMSDs become zero? Perhaps, the relation of the optimum product and the WA-average product is not completely clear (at least to me) and might deserve some clarifications in the text.

---

## Author Comment (AC1) · 21 Dec 2017

This study has addressed the difficulties in merging different ET products – in particular when the spatial resolutions are vastly different – those of coarse grid (25 km) versus in-situ observations (which are fetch dependent). The detailed description of the methodologies and the analysis of results as well as the discussion are very helpful to help the reader understand the challenges in such undertaking. The conclusions are honestly drawn based on the results. On the basis of the above facts, I recommend the publication at HESS after some minor revision.

R. We thank the reviewer for taking his/her time to review our paper, and we are glad to see that he/she thinks that the paper is suitable for publication at HESS.

[Figure]

Given the issues raised by the manuscript, which mentions issues like dependence of the products to be merged, the tower coverage, errors, and spatial representativeness of their measurements at the products resolution, and the nature of the ET product errors, I would suggest to use a more general title e.g. 'On issues in local tower-based merging of land evaporation products' or something similar. The current title is specific but I am not sure that the merged product is more useful than each of them individually and the true contribution of the study is to enlist and highlight these issues to the community.

R. We agree with the reviewer that the paper is more about the process of merging the products rather than about providing a successful merge product. Indeed, we do not claim that the merged product is a solid alternative to the individual products at this stage, though we hope that this will be the case when continuing this work with original estimates of finer resolution, derived with more independent forcings, and for more extended periods. We like the suggestion of the reviewer for a new title and we will change the title as suggested.

P8L24-25: I was not sure what 'a station-averaged square temporal correlation of 0.96.' – is this the coefficient of determination?

R. The 0.96 value was obtained by calculating first the Pearson correlation coefficient between the corrected and uncorrected fluxes at each station, squaring that value, and then averaging the individual station square correlations over all stations. We will rephrase as: "If the squared correlation coefficient between uncorrected and corrected fluxes is calculated at each station and then averaged over all stations, we obtain 0.96, showing that the uncorrected and corrected fluxes correlate very well in time.

P12L15: I was not sure what 'the satellite surface meteorology' refers to.

R. It refers to the inputs used by the ET models related to meteorological fields, in this case the surface radiation, the near-surface air temperature and humidity, and the precipitation. To make it clear we will rephrase as: "Bias can also be present between the

surface meteorological products used by the evaporation models, such as the surface radiation, or the near-surface air temperature and humidity, or the precipitation, and the real meteorological conditions at the tower".

---

## Author Comment (AC2) · 21 Dec 2017

The authors investigate the added value of merging two land ET products based on their performance with respect to tower-based ET. This is definitively an interesting topic in particular in the light of the existing large uncertainties in ET estimations. General comments and questions: The study is well written and provides interesting insights in the performance of the two used ET models. These seem to perform very similar and the merge of them does not provide a significant added value. I'm wondering thus if the use of other WACMOS- ET models with more diverse performance at the tower sites could be a better test case for the proposed merging procedure (instead of having two already similarly well- performing models with not much of room for improvement). Can the authors comment, why they did not include a more diverse

palette of models?

R. First, we would like to thank the reviewer for taking his/her time to review the paper.

We choose GLEAM and PTJPL to study a possible product merge as we thought that the more interesting challenge would be to merge the two project ET products showing more skills to capture tower fluxes and large-scale inferred evaporation. Being both already closer to the tower fluxes, we wanted to see if adding the tower information could result in a better ET product, which could have been of utility for the project. The reviewer is possibly right than merging with one of the less performing-models would have resulted in a more diverse performance at the towers and perhaps a more illustrative merging exercise, but as mentioned before, our main objective was to improved the performance of the two more skilled models.

In any case, as mentioned at the end of the paper, we plan to continue this with ET estimates of finer resolution, derived with more independent forcings, and for more extended periods. It is likely that the more independent model forcings will result in a more diverse performance at the towers. This, together with a larger pool of tower stations from the more extended period, could make the tower fluxes more informative, contributing more to the merged product. We will also be looking at adding other products, such as a newer global version of SEBS at 5km and daily resolution currently available from the SEBS developer team, or the new MODIS ET product version 6.

Also, how are data gaps in tower data and non-consistent temporal coverage of the towers treated? This might influence the analysis of the derived merging weights if the station sample changes over time.

R. There are certainly gaps in our processed tower data because, even if a station data record was complete, we remove the rainy intervals. We deal with this by requiring a minimum number of 20 daily observations in the running window selecting the time interval to derive the daily weights. Otherwise the weights will not be computed for that specific day. The latter happens in a very few occasions so it is not critical for the
results.

Regarding temporal coverage, for the 84 stations considered, 6, 14, 24, 9, and 31 stations had 2, 3, 4, 5, and 6 years of data, respectively. Clearly, the shorter the time period the less confidence we should have in the weights because the weights may be too specific to the climate conditions of these few years (e.g., especially dry or wet conditions). If these weights were then applied to merge the products at that location but for more standard conditions, the merge may not be optimal.

So, yes, there can be an impact due to data gaps and temporal coverage, but how this is influencing the analysis is difficult to judge. A possibility to minimize the impact of the temporal coverage is to keep only towers having data for a relatively large number of years, but that considerably reduces the already small number of towers. For instance, allowing at least 4 years of data removes half of the towers considered in the study.

Specific comments:
 page 3, line 12: What about other observational data? E.g. lysimeters or catchment wide estimates could be mentioned as well here.

R. We have used catchment wide estimates to evaluate our ET models during the WACMOS-ET project (Miralles et al., 2016), but we would possibly not call them observations, at least not ground observations. Lysimeters are definitely ground observations. However, as far as we know there is no organized network of lysimeter measurements, which may facilitate their widespread use for global evaluation of ET models. Nevertheless, we can mention them. We will rephrase as: "Ground measurements of land heat fluxes are typically conducted during field experiments (Pauwels et al., 2008) and by more permanent lysimeters (Hirsch et al., 2017) and flux tower networks (Baldocchi et al., 2001)".

page 5, lines 24/25: Do you expect an impact on the merging weighs when sub-daily simulations and tower data would be considered?

R. Yes, but only if the ET models skills were markedly different at different times of

the day. In that case the weights for specific times of the day would differ from the daily ones. However, from our sub-daily simulations presented in Michel et al. 2016 we did not notice very different skills at different times of the day when comparing with the tower data, and not much more skill was gained by producing daily ET based on 3-hourly input as opposed to forcing the models with the original daily input. We could then speculate that the weights will not be dramatically different in our case. In any case, we do not envisage to produce more sub-daily runs within this project given that, on the one hand, we do not have had many requests for sub-daily simulations estimates, and on the other hand, ET model updates are mostly focusing only on the daily scale.

page 7, line 18: Obsolete brackets between the two cited papers. 

R. Thanks, we will remove them.

page 8, line 1: What happens if the two data sources for precipitation disagree? 

R. Certainly there are moments when the precipitation at the tower disagrees with the gridded precipitation, which is expected due to the different spatial resolutions and the unavoidable errors associated to the gridded product. We only leave days when there was no rain from the gridded product and the tower recordings (for the towers where precipitation was measured). We will rephrase to make it clearer as: "(2) masking measurements for rainy intervals, only leaving observations if both the global precipitation product and the local measurements (if available) do not indicate precipitation (eddy-covariance measurements are generally less reliable during precipitation events)".

page 12, line 8, "estimated over the time series of available errors": What about differences in the length of the EC time series or differences in the occurrence of data gaps between the towers? How is this taken into account in the analysis of the weights?

R. As we explained above, all the towers do not have the same number of years of data, and there are data gaps, especially because we filter for precipitating conditions.

This could have an impact in the weights analysis, especially if for the stations with a very short number of years, the existing years are not representative of the typical climate conditions, but we cannot quantify the impact of this. Being much stricter with the data gaps and number of years was not possible here, as the number if towers and weights would have been considerably reduced, as previously discussed.

We will add the information about the number of years in the text: "This processing selects 84 stations for the 2002-2007 study period. Notice that not all stations completely cover this period, with 6, 14, 24, 9, and 31 stations having 2, 3, 4, 5, and 6 years of data, respectively. Their geographical locations . . .".

We will also comment on the possible impact of a relatively short number of years to derive the weights when describing Figure 7, as the US-SRM station shows large inter-annual variability. We will add: "Figure 7 showed that at some stations inter-annual variability is large. At these stations, if only a relatively short number of years exist to derive the weights, the weights may not be representative of the overall climate conditions at the tower. For instance, for the US-SRM site, weights derived by only considering the year 2005 or the year 2007 would have possibly been quite different. This suggests that there can be an impact regarding the different number of years available per station, and that more confidence should be placed in the derived weights the longer the time period available at the tower. Limiting the study to stations with a relatively large number of years can be used to minimize the impact of this, but in our case that severely reduces the number of towers. For instance, if we only derive weights if at least 4 years of data are available, half of the towers would have been removed."

page 13, lines 3-5: Rephrase: "A 61-day running window was found to provide ..."

R. We will rephrase like that.

page 13, lines 3-5: Is there a minimum requirement of data availability for deriving the weight within the window?

R. Yes, 20 daily observations as mentioned before. We will add this to the text as: "A number of 30 days before and after each calendar day was found to provide a good compromise between the smoothness of weights and the number of samples required, so a 61-day running window was found to provide the daily weights.
 Notice that due to the masking of the tower data at very few occasions the 61 daily estimates are present in the running window, and 20 daily values reasonably spread in the running window are required as a minimum number of ET estimates to derive the weights".

page 13, lines 9-13: Sounds a bit confusing and not so clear (at this point of the paper at least). Try to re-formulate being a bit more specific (reasons to believe that?).

R. We are just stating here the impossibility to evaluate interception as we mask the data record for precipitation conditions. We will rephrase trying to be clearer as: "As the tower measurements are masked for rainy intervals, we cannot evaluate the interception loss of the modelled ET. Therefore, only the sum of soil evaporation and transpiration is compared with the tower data and weighted. This is clearly a limitation as in most cases the total ET, and not only the soil evaporation and transpiration, are required. This means that an estimate of interception loss should also be provided, either by assuming that GLEAM and PT-JPL interception loss are equally uncertain and adding their average to the weighted soil evaporation and transpiration, or by adding just one of the GLEAM or PT-JPL individual interception losses, if there are reasons to believe that one is overall more certain than the other.
 page 14, lines 14-20, "optimum product": How often is the optimum product one of the two models (i.e., weight of one)? From Fig. 5 it looks like a weight of one is never reached.

R. On average over all stations, 80% of the times, so quite often. Or in other words, on average over all stations only once every 5 days the tower ET is between the GLEAM and PT-JPL estimates, and the optimum product is the tower ET. But notice that this is not related to the weights of Figure 5. The reviewer is possibly confusing the weights derivation with the construction of the optimum product. See please our response to your last comment, where we clarify the relation between weights and the optimum

product.

page 19, Fig. 4: The legend interferes with the figure information, please increase the y-axis range a bit.

R. Sure, we will fix that.

page 19, Fig. 4 caption: Is it the optimum product you are comparing with the tower ET, not the WA-product? From page 14, I get that the optimum product is either the tower ET or one of the two models. Assuming that it's often the tower ET (judging from Fig. 5 where weights never reach one), shouldn't the RMSDs become zero? Perhaps, the relation of the optimum product and the WA-average product is not completely clear (at least to me) and might deserve some clarifications in the text.

R. Yes, it is the optimum product. We will try to clarify the differences between the weights of the merge product and the construction of the optimum product in the following lines.

The weights of the merge product are derived by calculating the variance of the error distribution over a time interval, as described in Section 2.3. For the weights to be close to one the error variance of one of the models has to be much smaller than the other one, which is seldom the case. Maximum values tend to be around 0.9, as it is illustrated in Figure 6. This means that the merge product is rarely just one of the GLEAM or PT-JPL estimates.

Now, the optimum product is not derived by applying those weights, but by comparing tower and model estimates at every day, as also described in Section 2.3. If the tower ET is between the GLEAM and PT-JPL estimates, the tower ET is then the optimum estimate, but if both the GLEAM and PT-JPL estimates are below or above the tower ET, the closest model ET is the optimum estimate. So this is done independent of any weight estimation, and it is not related to the weights presented in Figure 5, which are the weights of the merge product derived by using the error variances.

For the RMSD of the optimum product to be zero at a given tower, that would imply that the tower ET is always between the GLEAM and PT-JPL estimates. This is never the case for the 84 considered stations.

We try to make this clearer this by rephrasing the description of the optimum product as: " When improving a product and comparing with a reference, the common targets are correlation unity and zero RMSD. Here, instead, we define a product that minimizes the RMSD with the tower ET, and call it, in the context of our merging strategy, the optimum product. At the days when the tower ET is between the GLEAM and PT-JPL estimates, the tower ET is the optimum estimate. But there is also the possibility that both GLEAM and PT-JPL estimates are below or above the tower ET. For the 84 towers considered here this is the most common situation, happening on average over all stations for around 80% of the days. In this case the merge product will never be the tower ET because, independent of the weight values assigned by the error variance analysis, the merge product is bound by the original model ET estimates. This is precisely the reason why correlation unity and zero RMSD can never be achieved here. For the optimum product, the closest model ET to the tower ET is the value that minimizes the RMSD with the tower ET, and this is the value selected for the optimum product in this case".

---

## Referee Comment (RC3) · Anonymous Referee #3 · 28 Dec 2017

This manuscript describes work to combine two ET products, PT-JPL and GLEAM, using a weighted average, with weight determined by fit to tower observations. The resulting product is limited to the locations of the towers, and no attempt is made at extrapolation to other sites. While the manuscript is well written, and the analysis sound, the work itself is not well motivated and, as currently presented, does not represent a significant contribution.

The merged product presented in this manuscript does not add any value to the ET products that are already available. The motivation seems to be to merge the two ET products (PT-JPL and GLEAM) to produce a new product that is as close to the tower ET time series as possible. How is this new merged product then any more useful than the original tower ET time series? The merged ET product has not been independently

[Figure]

Creative Commons CC BY license logo

evaluated. It is shown to be closer to the tower observations, but this is by design. Given that the tower observations also have errors (and given how closely the merged product has been fit to the tower obs), it does not follow that the merged product is necessarily more accurate. I am concerned that the merged product is over-fitting to the tower obs (weights calculated independently at each location, using a moving temporal window)

The work is not very well motivated. Why merge just these two products? Why not merge as many as are available, or as many as meet some pre-defined standard? The selection of just these two products is particularly awkward given that they are not independent.

To be publishable this work must i) provide a product that adds value in some way to the original products., and ii) the resulting data set must also be independently verified. The most obvious way to achieve this would be to spatially extrapolate the weightings. This could potentially provide a new product with (near-) global coverage that is more accurate than either of the original gridded ET data sets, and would also allow independent verification against withheld tower observations.

If this is not possible, I suggest that the manuscript be re-submitted and re-written (with additional discussion and conclusions) to focus on evaluating the GLEAM and PT-JPL products against tower obs.

MINOR COMMENTS:

Section 2: There is not enough information here for the reader to understand how the two products are calculated and what their main differences are. Please provide full details of the methodology of each product, rather than relying on previous work.

P5, L24: give the specific resolutions.

P8, L5: mention that the station coverage is not globally uniform, with nearly all stations in Europe and the US.

P8, L20: 'corrected fluxes are preferred". Provide citation. Also, for the results provided in this paragraph for the corrected fluxes, how were they corrected?

Equation 1: add a sentence to describe what this metric is measuring (something like "the first term is the mismatch between the land cover at the tower and at the grid cell level, and the remaining terms are the net mismatch in land cover types across the two resolutions").

P14, paragraph from line 10: the text here implies that the motivation is to match the tower ET as closely as possible, but the tower ET will also include errors. This paragraph should be re-written to acknowledge that the tower ET will also include errors (and the methodology perhaps adjusted to not over-fit to the ET data)

P15, L10. The use of the full seasonal cycle concerns me. In general, different ET products agree reasonably well in terms of the seasonal cycle (Jimenez et al. 2011; Mueller et al. 2011; Miralles et al. 2011). It is the anomalies that have more dis-agreement, and should then be the focus of efforts to improve / combine ET products. Also, using anomalies would be consistent with the assumption in the methodology that there are no biases. The reason given for not using anomalies is that there is insufficient tower data - if there really is insufficient data, this implies that ET cannot be trained on tower obs.

Jimenez, C., and Coauthors, 2011: Global intercomparison of 12 land surface heat flux estimates. Journal of Geophysical Research: Atmospheres, 116, D02 102, doi:10.1029/2010JD014545. Miralles, D., T. Holmes, R. de Jeu, J. Gash, A. Meesters, and A. Dolman, 2011: Global land- surface evaporation estimated from satellite-based observations. Hydrology and Earth System Sciences, 15, 453–469, doi:10.5194/hess-15-453-2011. Mueller, B., and Coauthors, 2011: Evaluation of global observations-based evapotranspiration datasets and IPCC AR4 simulations. Geophysical Research Letters, 38, L06 402, doi:10.1029/ 2010GL046230.

P18, L18: EC is known to under-estimate the fluxes. Using the sum of LH and SH as

the incoming energy will almost certainly give an underestimate.

Figure 5: what is causing the sudden changes in the time series? The 91 day windows used shouldn't suddenly change like this.

Figure 7: This sudden increase in the tower ET in the upper panels look incorrect (and seem to occur at the same time each year - unless these are preceded by significant rain events, this don't look right). This time series needs to be checked, carefully QC-ed, and unusual features like this should be explained in the text.

The work would benefit from being placed within the context of other efforts to estimate ET with tower data / statistical methods. In particular the MTE product should be mentioned somewhere, as an example of using tower EC obs to estimate global ET.

Jung, M., M. Reichstein, and A. Bondeau, 2009: Towards global empirical upscaling of FLUXNET eddy covariance observations: validation of a model tree ensemble approach using a biosphere model. Biogeosciences, 6, 2001–2013, doi:10.5194/bg-6-2001-2009.

---

## Author Comment (AC3) · 8 Jan 2018

**Reviewer 3**

This manuscript describes work to combine two ET products, PT-JPL and GLEAM, using a weighted average, with weight determined by fit to tower observations. The resulting product is limited to the locations of the towers, and no attempt is made at extrapolation to other sites. While the manuscript is well written, and the analysis sound, the work itself is not well motivated and, as currently presented, does not represent a significant contribution.

*R. We thank the reviewer for taking his/her time for a detailed review of our paper, but we certainly disagree about his/her rating of our paper. We also clarify that the resulting merging method is a first and necessary step towards a global merger that is certainly envisaged.*

The merged product presented in this manuscript does not add any value to the ET products that are already available. The motivation seems to be to merge the two ET products (PT-JPL and GLEAM) to produce a new product that is as close to the tower ET time series as possible. How is this new merged product then any more useful than the original tower ET time series?

*R. The local merge of GLEAM and PT-JPL at the selected towers was the first step to produce a merge product. Perhaps we were not clear in the motivation and objectives. We fully agree with the reviewer that the merge product would be useful outside the locations of the towers, but not where we already have the tower estimates. But the first step is to prove that the merge product fits the tower data better than the individual products at the tower sites, and this is mainly what this paper is about. It is not an obvious exercise, as the tests carried out in the paper show, and we definitely think that it is worth publishing.*

The merged ET product has not been independently evaluated. It is shown to be closer to the tower observations, but this is by design. Given that the tower observations also have errors (and given how closely the merged product has been fit to the tower obs), it does not follow that the merged product is necessarily more accurate. I am concerned that the merged product is over-fitting to the tower obs (weights calculated independently at each location, using a moving temporal window).

*R. Over-fitting is always a concern with these statistical approaches, but for the moment we are just trying to show that at each specific site the optimal*

*estimator can result in a product better fitting the tower ET. Now, for the global merger over-fitting definitivel needs to be tested. If the merge product were doing well at a specific location, but poorly in a similar but different location, then we would have over-fitting, or poor generalization issuess. The usual approaches to test this could then be applied, such as the cross validation techniques where stations are grouped by land cover and each land cover dataset stratified in independent parts to derive the weights and test the weight performance. Still, these are in a way "self-contained" tests and not independent evaluations, as we keep using tower data, and we can only prove that the merged product is more "accurate" with respect to the tower data. At the individual site scale, we do not see how else the merge product accuracy can be tested, and we would be happy to hear suggestions from the reviewer in this sense. If spatial integrations are allowed, then more "independent" evaluations can be carried out by comparing with "inferred" ET.*

*Certainly the tower ET also has errors, as described in the paper, and any methodology that tries to fit to the tower ET is likely to inherit those. Nevertheless, there is some consensus in the ET community that the tower fluxes are our best shot for ground "truth" at ecosystem scale. The optimal linear estimator applied here tries to minimize the error variance of the merged product with respect to a reference, in this case the tower observations, and in that sense certainly by definition the merge product tries to get closer to the tower observations, compared with the original ET estimates.*

The work is not very well motivated. Why merge just these two products? Why not merge as many as are available, or as many as meet some pre-defined standard? The selection of just these two products is particularly awkward given that they are not independent.

*R. We stated in the Introduction the reasons behind merging GLEAM and PT-JPL. In short, after years of testing different methodologies to derive "satellite-based" ET products, GLEAM and PT-JPL showed more skills than others tested methodologies (Michel et al., 2016, Miralles et al., 2016, McCabbe et al., 2016), so we wanted to see if we could merge them to produce a better product. We think that this is a legitimate objective in the framework of our WACMOS-ET project and connected initiatives, such as the GEWEX LandFlux initiative (https://halo.kaust.edu.sa/Pages/GEWEX_Landflux.aspx), and we do not see anything awkward here even if the products are not completely independent. So, although merging a large number of products is also a valid objective (e.g., the*

*recent Yao et al, 2017), this is not our objective in this study.*

To be publishable this work must i) provide a product that adds value in some way to the original products., and ii) the resulting data set must also be independently verified.

*R. We disagree that research on this topic can only be published if it results in a new product. We believe that what we learned about merging these two specific products is of broad interest for other colleagues working on these topics, even if it is just a firs step for a successful merger.*

The most obvious way to achieve this would be to spatially extrapolate the weightings. This could potentially provide a new product with (near-) global coverage that is more accurate than either of the original gridded ET data sets, and would also allow independent verification against withheld tower observations.

*R. Extrapolating the weights is certainly required to produce a global merger. It was already mention in the Conclusions, and it is something we have already worked on. Although we do not think that this specific merge was distinctively better than the simple average of GLEAM and PT-JPL, we already did some tests with the current weights in preparation for future merging efforts. So far we have used a multilayer-perceptron driven with a selection of the ET model inputs and trained to reproduce the current 84 station weights. This first setup seemed to show some skills to extrapolate the weights globally, with the predicted weights averaged over all stations correlating 0.7 with the original weights.*

*A figure showing preliminary globally seasonally averaged weights is displayed below, followed by a second figure showing seasonally averaged ET differences between the global average of GLEAM and PTJPL and the global product derived from applying the globally extrapolated weights, normalized by the seasonal average ET. Some clear seasonal and geographical patterns are observed in the extrapolated global weights and ET differences, which, as the reviewer indicated, would need to be assessed by confronting the global merge product with independent estimates.*

[Figure]

*Figure A. Global seasonally averaged weights. GLEAM weighted more than PT-JPL is indicated in red. PT-JPL weighted more than GLEAM is indicated in blue. The seasonal weight is the value shown in the colour bar plus 0.5.*

[Figure]

*Figure B. Seasonally averaged E differences between the simple-average and the weighted product normalized by the seasonal average E.*

If this is not possible, I suggest that the manuscript be re-submitted and re-written (with additional discussion and conclusions) to focus on evaluating the GLEAM and PT-JPL products against tower obs.

*R. We have already published GLEAM and PT-JPL evaluations against tower observations and independent ET products, both at the tower and global scale, with the model using a variety of forcing datasets (Michel et al., 2016, Miralles et al., 2016, McCabe et al., 2016). For us, this is a step further in a quest for a global merger of these types of ET methodologies, and we firmly believe that reporting these our first findings will be well received by interested colleagues.*

*Our next attempt to provide a more successful merge product will be based on using GLEAM and PT-JPL run with less common forcings, and for more extended periods. It is likely that the more independent model forcings will result in a more diverse performance at the towers. This, together with a larger pool of tower stations from the more extended period, could make the tower fluxes more informative, contributing more to the merged product. We will also be looking at adding ET estimates from the other two models run by WACMOS-ET, such as a newer global version of SEBS at 5km and daily resolution currently available from the SEBS developer team, or the new MODIS ET product version 6. We hope that a more extended dataset of tower locations and the more diverse weights of the future merge will be working on will also help to improve the global extrapolation of the weights. This will be reported in our next paper, together with independent evaluations of the merged product by comparing with other ET products and with inferred E derived from long time averages of basin precipitation and river runoff, as we have done in Miralles et al., 2016.*

*To address the reviewer concerns, we will make the following changes to the original manuscript:*

*(1) As suggested by another reviewer, we will change the title to make clear that the paper is more about the process of merging the products rather than about providing a successful merge product. We are considering different options, e.g., "Exploring the merging of two global land evaporation datasets based on local measurements".*

*(2) We will add to the literature review the statistical approaches that regress directly the tower ET on explanatory variables: "These efforts range from simply averaging a number of ET products (Mueller et al., 2003) to more complex approaches, such as fusion algorithms where the original ET products are combined to reproduce flux observations (Yao et al., :2017), or integration methodologies that seek consistency between ET products and related products*

*of the water cycle (Aires, 2014, Munier et al., 2014). Notice that there are also ET products based on a direct regression of tower ET on a set of explanatory variables (Jung et al., 2009, Wang et al., 2010)".*

*(3) We will focus more the paper objectives by adding: "Nevertheless, substantial differences were found between both model products. As such, we pose the question: can a product combining the GLEAM and PT-JPL estimates result in a more accurate ET estimate? We start here by investigating a weighted combination of GLEAM and PT-JPL estimates at some selected locations. Ideally, the weight assigned to each product during their merging should be based on an accurate description of the specific product uncertainties. However, even if some attempts to derive model uncertainty exist (Miralles et al., 2011a; Badgley et al., 2015; Loew et al., 2016), the complexity to derive estimates of ET from remote sensing data means that reliable quality assessment is only attained through validation against tower flux measurements. Therefore, here we propose a flux tower-based weighting of GLEAM and PT-JPL and an investigation of the performance of the resulting merger over a selection of tower sites".*

*(4) At the end of the section describing the merging technique we will add: "It should be noticed that the merging technique described would also need weights derivation outside the tower locations for the merge product to be useful. This will require an extrapolation technique, which can potentially be based on a regression of the weights on some explanatory variables. Likely candidates for the latter can be the inputs used to drive the ET models, assuming that relationships between the weights and the model inputs exist. Here we limit our study to investigate the performance of the merging technique at the tower sites, but this needs to be addressed in forthcoming efforts to provide merged estimates outside the location where the weights are derived.*

MINOR COMMENTS:

Section 2: There is not enough information here for the reader to understand how the two products are calculated and what their main differences are. Please provide full details of the methodology of each product, rather than relying on previous work.

*R. This is the third paper of the WACMOS-ET project, the first two ones also published in this journal. The GLEAM and PT-JPL models were described in more detailed, including their main equations, in the first paper, while for the second and this third one we only describe the main characteristics of the models. We are certain that any reader interested in this work would need to glance through the previous papers to follow this one, so we are not sure that fully describing the models here will be that useful. The same applies to the model forcings, which we described in detail in the first paper, and that we only summarized in the second and this third paper. We will consult with the editor about this, as we already had plagiarism complains precisely by mentioning again in this paper project elements already described in the first papers.*

P5, L24: give the specific resolutions.

*R. We will rephrase as: "Notice that the WACMOS-ET runs were done at 3-hourly and daily time resolutions, while only daily estimates are calculated for this study".*

P8, L5: mention that the station coverage is not globally uniform, with nearly all stations in Europe and the US.

*R. We will rephrase as: "This processing selects 84 stations for the 2002-2007 study period, with nearly all stations in Europe and US".*

P8, L20: 'corrected fluxes are preferred". Provide citation. Also, for the results provided in this paragraph for the corrected fluxes, how were they corrected?

*R. The citation is Foken et al., 2006, already provided. We will rephrase as:*

*"Techniques to correct this exist (Foken et al., 2006), and corrected fluxes applying these techniques are typically preferred over the original uncorrected observations."*

Equation 1: add a sentence to describe what this metric is measuring (something like "the first term is the mismatch between the land cover at the tower and at the grid cell level, and the remaining terms are the net mismatch in land cover types across the two resolutions").

*R. We will add as suggested: " ...where the tower is situated. The first term is the mismatch between the land cover at the tower and at the grid cell level, and the remaining terms are the net mismatch in land cover types across the two resolutions. It takes the value ...".*

P14, paragraph from line 10: the text here implies that the motivation is to match the tower ET as closely as possible, but the tower ET will also include errors. This paragraph should be re-written to acknowledge that the tower ET will also include errors (and the methodology perhaps adjusted to not over-fit to the ET data)

*R. We will improve this paragraph to make it clearer and add a note concerning the tower errors at the end: "When improving a product and comparing with a reference, the common targets are correlation unity and zero RMSD. Here, instead, we define a product that minimizes the RMSD with the tower ET, and refer to it, in the context of our merging strategy, the optimum product. At the days when the tower-measured ET is between the GLEAM and PT-JPL estimates, the optimum estimate equals the tower-measured ET. However, when both GLEAM and PT-JPL estimates are below or above the tower ET, the merge product will never be the tower-measured ET because it is bounded by the two model estimates. This is the main reason why correlation unity and zero RMSD can never be achieved here. For the optimum product, the closest model ET to the tower ET is the value that minimizes the RMSD with the tower ET, and this is the value selected for the optimum product in this case. For the 84 towers considered here this is the most common situation, happening on average over*

*all stations for around 80% of the days. In addition, as discussed in Section 2.2.2, the tower ET is also subject to errors, so the optimum product will inherit those errors at the instants where it takes the tower value. Therefore, there is not claim that the optimum product at those instants represents the unknown true ET, but the ET at the tower footprint as measured by the eddy-covariance instruments".*

P15, L10. The use of the full seasonal cycle concerns me. In general, different ET products agree reasonably well in terms of the seasonal cycle (Jimenez et al. 2011; Mueller et al. 2011; Miralles et al. 2011). It is the anomalies that have more disagreement, and should then be the focus of efforts to improve / combine ET products. Also, using anomalies would be consistent with the assumption in the methodology that there are no biases. The reason given for not using anomalies is that there is insufficient tower data - if there really is insufficient data, this implies that ET cannot be trained on tower obs.

Jimenez, C., and Coauthors, 2011: Global intercomparison of 12 land surface heat flux estimates. Journal of Geophysical Research: Atmospheres, 116, D02 102, doi:10.1029/2010JD014545. Miralles, D., T. Holmes, R. de Jeu, J. Gash, A. Meesters, and A. Dolman, 2011: Global land- surface evaporation estimated from satellite-based observations. Hydrology and Earth System Sciences, 15, 453–469, doi:10.5194/hess- 15-453-2011. Mueller, B., and Coauthors, 2011: Evaluation of global observations- based evapotranspiration datasets and IPCC AR4 simulations. Geophysical Research Letters, 38, L06 402, doi:10.1029/ 2010GL046230.

*R. The better agreement of ET products when the full seasonal cycle is considered is just the result of correlating two variables with marked lows and highs. In general, the more pronounced the seasonal cycle, the better the agreement in terms of correlation. At locations when the seasonal cycle is smaller, such as tropical forests, the agreement of the absolute ET values is much poorer in terms of correlation. This is not exclusive of ET estimates, it is also the case for other variable with strong seasonal cycles (e.g., radiation, temperature, precipitation).*

*Certainly, working with the anomalies would have been interesting, but this cannot be reliably achieved with our present data records. To work with*

*anomalies, a robust calculation of the seasonal cycle at the tower locations is needed. How many years would be acceptable? If we take the whole 1980-2015 FLUXNET2015 synthesis data set, and we conserve stations having at least 10 years of data, we are left with ~25% of the stations. If we take 5 years, which can be disputed as a sufficient number of years for a climatology, we still remove ~50% of the stations. As mentioned by the reviewer, the tower dataset is already severely limited in terms of geographical coverage, so such dramatic cuts in the number of stations is not very helpful for any merging methodology.*

*As mentioned in the paper, even if we do not work with anomalies, we try to show seasonal agreement between the different products, not just annual values, to reduce somehow the effect of the seasonal cycle in the analyses. Also, as the weights are calculated over a 61-day running window, there will be times of the year where the inter-seasonality within the 61-day running windows is small. At least for those times of the year there should not be much difference between the weights working in an absolute or anomaly space.*

*Nevertheless, independent of all this measures to mitigate the impacts of the seasonal cycle, we are certainly not working in an anomaly space. But for us, stating that "this implies that ET cannot be trained in tower observations" is not justified. It is still a challenge to reproduce the absolute ET values, as shown in the references given by the reviewer, or Figures 2 and 3 for this particular case, and as far as we can see most ET product merging efforts based on tower data work with absolute values.*

P18, L18: EC is known to under-estimate the fluxes. Using the sum of LH and SH as the incoming energy will almost certainly give an underestimate.

*R. True. We have been very clear about it in the same paragraph, stating the 6.1% underestimation when averaged over all the stations. We would argue than this underestimation has a smaller impact here, compared with other statistical approaches directly targeting the tower ET, such as the MTE product suggested later on by the reviewer. This is because we are not directly reproducing the tower ET, but weighting the original GLEAM and PT estimates. There can be an effect when deriving the error variance, as the relative differences of GLEAM and PT-JPL with the tower ET can change if corrected or uncorrected tower fluxes are used, but the merge product still remains bound by the original estimates.*

Figure 5: what is causing the sudden changes in the time series? The 91 day windows used shouldn't suddenly change like this.

*R. The reviewer is right, and continuous 61-day windows should not produce abrupt changes in this plot. However, at some locations the weights do not exist for all days. This happens at a few stations, as we impose that there should be at least 20 well spread daily values in the 61-day running window to derive the weights. For instance, in the right panel of Figure 5 the maximum values before the sudden decrease at day 180 correspond to the station CN-Dan. Due to observations quality and rain episodes there are not enough daily values to derive the weights for this station for the next few days, and the next maximum value comes from a new station with a lower value, producing the discontinuity.*

*We will be adding to the text describing the weight calculation: "A number of 30 days before and after each calendar day was found to provide a good compromise between the smoothness of weights and the number of samples required, so a 61-day running window was used to provide the daily weights. Notice that due to the masking of the tower data at very few occasions the 61 daily estimates are present in the running window, and 20 daily values reasonably spread in the running window are required as a minimum number of ET estimates to derive the weights. At most stations weight values exist for nearly all days, but at 8 stations there are larger gaps. The worst case is the tropical BR-Sa3 station, where the frequent rainy episodes complicate the derivation of the weights."*

*We will also replace the maximum and minimum curves in the Figure with the 5% and 95% percentiles, which are very close to the maximum and minimum values, but less sensitive to the discontinuities caused by the gaps in the time series of the weights at a few stations. The new figure can be found below.*

[Figure]

Figure 5. Daily statistics of weights over all forest (left), shrub/savanna (middle), and crop/grass (right) sites, plotted as the difference of the absolute weights and 0.5 (referred to as weight devia- tions, Δweight). A GLEAM Δweight of a means that GLEAM ET is weighted 0.5 + a, and PT-JPL ET 0.5 − a, while a PT-JPL Δweight of b means that GLEAM ET is weighted 0.5 − b, and PT-JPL ET 0.5 + b. Displayed are the mean, and the 5%, 25%, 75% and 95% percentiles.

Figure 7: This sudden increase in the tower ET in the upper panels look incorrect (and seem to occur at the same time each year - unless these are preceded by significant rain events, this don't look right). This time series needs to be checked, carefully QC- ed, and unusual features like this should be explained in the text.

*R. As described in Section 2.2.2, the tower data was quality-controlled using the provided quality flags, and the represented fluxes were not marked as problematic. This site is a semi-arid savannah where vegetation development and associated fluxes are tightly linked to precipitation and humidity conditions. Station precipitation and soil moisture measurements in the 2004-2007 period can be found in Scott et al., 2009 (J. Geophys. Res., 114, G04004, doi:10.1029/2008JG000900), and match the general behaviour of the fluxes.*

*However, we plotted the ET estimates used for the derivation of the weights with the rainy episodes removed. This together with the running window used to smooth the lines produced the abrupt changes at the arrival of the summer rainfall, when many ET estimate are removed to derive the weights. To remove any confusion, we will be re-plotting the full time series, with the rainy days*

*marked also in the figure. We can have the merge product for all days, as the weights exist for all days, although to study their agreement with the tower ET we only consider the non-rainy days, as discussed in the article. The new figure can be found below.*

[Figure]

Figure 7. *ET time series at the sites US-SRM (top) and US-Nei (bottom). The daily values are time smoothed with a 10-day moving averaged window to better display the more persistent temporal features. Rainfall is marked with black circles in the x-axis, showing dry and rainy periods for US-SRM, and a more evenly distributed rain along the year at US-Nei.*

*We will also modify the text to reflect these changes: "At the semi-arid grassland US-SRM site there are relatively large ET seasonal variability, with the ET tightly linked to the precipitation and associated increased in soil moisture (Scott et al., 2019). In terms of correlation, GLEAM and PT-JPL do not agree very well with the tower ET, with correlations of 0.31 and 0.24 for GLEAM and PT-JPL, respectively, calculated over the non-rainy days. Correlations are higher when calculated over all days, with values of 0.81 and 0.74 for GLEAM and PT-JPL, respectively, but as discussed in Section 2.2.2, we remove rainy episodes when analysing the data. GLEAM seems to capture better the spring ET decrease associated with the increase in temperatures and decrease in soil moisture, and both GLEAM and PT-JPL capture well the sudden increase in ET values at the beginning of summer related to the rainfall coming*

*from the North American monsoon. In this case, the larger weighting for GLEAM results in a weighted product that seems closer to the observed values, although the 0.37 correlation of the SA-merged product is not significantly higher than the 0.31 correlation of the WA-merged product".*

The work would benefit from being placed within the context of other efforts to estimate ET with tower data / statistical methods. In particular the MTE product should be mentioned somewhere, as an example of using tower EC obs to estimate global ET.

Jung, M., M. Reichstein, and A. Bondeau, 2009: Towards global empirical upscal- ing of FLUXNET eddy covariance observations: validation of a model tree ensemble approach using a biosphere model. Biogeosciences, 6, 2001–2013, doi:10.5194/bg-6- 2001-2009.

R. *We are very familiar with the MTE product, and compared the WACMOS-ET estimates with the MTE product in e.g. Miralles et al., 2016. This product is now cited as discussed above.*

---

## Author Comment (AC4) · 26 Jan 2018

We are adding some further comments to our previous response. As we hinted there, we were already working in the direction of (1) global extrapolation of the weights, (2) deriving global estimates, and (3) independently validating the new estimates. We have made sufficient progress in the last weeks that we feel now confident to include this work in a revised version of the paper, allowing us to expand the current work to tackle the main concerns of the reviewer.

In short, we summarize here the reviewer major concerns and our proposed actions to address them:

1. Why are these two products merged and not a large number of ET-products?

[Figure]

R. We will add the third model run globally during the WACMOS-ET project to the merged product to make the merging exercise more diverse in terms of models: The MODIS ET model. To our knowledge there are no other global, daily, publicly available satellite-based products covering the study period that could be added at this moment.

2. The energy balance closure error of the EC-data is an important issue not sufficiently addressed.

R. We will be comparing model performance at towers with and without energy balance closure, to assess the impact of this misbalance. This issue will be discussed in the revised manuscript. We will also perform a new evaluation of the FLUXNET2015 archive to assess whether new towers can be incorporated to the analyses.

3. Only estimates are provided at the tower locations, so the additional value of the product is also unclear.

R. We will be exploring the global extrapolation of the weights and deriving the globally merged estimates. This will be presented and discussed in the revised paper.

4. The merged ET-product is not independently validated. The towers ET-data also have considerable systematic and random errors. The fact that the merged ET-product fits the tower ET-data closer is by construction.

R. At ecosystem scale we will be applying bootstrapping techniques, stratifying the tower database to discuss the general validity of the weights. In addition, at basin scale, we will be comparing the original and merged estimates with inferred evaporation derived from precipitation and river runoff datasets. This analysis will be incorporated to the revised manuscript.

---

## Author Response (AR1)

Dear Editor,

Find here a point-by-point response to the reviews and a marked-up manuscript version. We have substantially changed the original manuscript to accommodate the demands from the three reviewers. The major changes are: (1) assessing the tower data set in terms of representing different biome and climate conditions; (2) producing global weights and the associated global product by extrapolating the local weights to the global landscape, and; (3) evaluating the global merged product by comparisons with inferred evaporation derived from basin-integrated precipitation and river run-off. To guide the reviewers through the manuscript we have marked it up with red sentences describing the changes.

Sincerely yours,

Carlos Jimenez (on behalf of all co-authors)

Review 1

This study has addressed the difficulties in merging different ET products – in particular when the spatial resolutions are vastly different – those of coarse grid (25 km) versus in-situ observations (which are fetch dependent).

The detailed description of the methodologies and the analysis of results as well as the discussion are very helpful to help the reader understand the challenges in such undertaking. The conclusions are honestly drawn based on the results.

On the basis of the above facts, I recommend the publication at HESS after some minor revision.

*R. We thank the reviewer for taking his/her time to review our paper, and we are glad to see that he/she thinks that the paper is suitable for publication at HESS. Following the advice of the third reviewer and editor, we have greatly revised the manuscript to include a global extrapolation of the weights we were already working on.*

Given the issues raised by the manuscript, which mentions issues like

dependence of the products to be merged, the tower coverage, errors, and spatial representativeness of their measurements at the products resolution, and the nature of the ET product errors, I would suggest to use a more general title e.g. 'On issues in local tower-based merging of land evaporation products' or something similar. The current title is specific but I am not sure that the merged product is more useful than each of them individually and the true contribution of the study is to enlist and highlight these issues to the community.

*R. We agree with the reviewer that the paper is more about the process of merging the products rather than about providing a successful merge product. Indeed, we do not claim that the merged product is a solid alternative to the individual products, even with he revised manuscript. The suggestion of the reviewer for a new title is appropriate and we changed along those lines to: "Exploring the merging of the global land evaporation WACMOS-ET products based on local tower measurements".*

P8L24-25: I was not sure what 'a station-averaged square temporal correlation of 0.96.' – is this the coefficient of determination?

*R. The 0.96 value was obtained by calculating first the Pearson correlation coefficient between the corrected and uncorrected fluxes at each station, squaring that value, and then averaging the individual station square correlations over all stations. We will rephrase as: "If the squared correlation coefficient between uncorrected and corrected fluxes is calculated at each station and then averaged over all stations, we obtain 0.96, showing that the uncorrected and corrected fluxes correlate very well in time.*

P12L15: I was not sure what 'the satellite surface meteorology' refers to.

*R. It refers to the inputs used by the ET models related to meteorological fields, in this case the surface radiation, the near-surface air temperature and humidity, and the precipitation. To make it clear we will rephrase as: "Bias can also be present between the surface meteorological products used by the evaporation models, such as the surface radiation, or the near-surface air temperature and humidity, and the real meteorological conditions at the tower".*

Review 2

The authors investigate the added value of merging two land ET products based

on their performance with respect to tower-based ET. This is definitively an interesting topic in particular in the light of the existing large uncertainties in ET estimations.

General comments and questions:The study is well written and provides interesting insights in the performance of the two used ET models. These seem to perform very similar and the merge of them does not provide a significant added value. I'm wondering thus if the use of other WACMOS- ET models with more diverse performance at the tower sites could be a better test case for the proposed merging procedure (instead of having two already similarly well-performing models with not much of room for improvement). Can the authors comment, why they did not include a more diverse palette of models?

*R. First, we would like to thank the reviewer for taking his/her time to review the paper.*

*We choose GLEAM and PT-JPL to study a possible product merge as we thought that the more interesting challenge would be to merge the two project ET products showing more skills to capture tower fluxes and large-scale inferred evaporation. Being both already closer to the tower fluxes, we wanted to see if adding the tower information could result in a better ET product, which could have been of utility for the project. Bit the reviewer is possibly right than merging with one of the less performing-models would have resulted in a more diverse performance at the towers, and perhaps a more illustrative merging exercise, so we added the third model rung globally during the project to the revised manuscript. Following the advice of the third reviewer and editor, we also include now a global extrapolation of the weights we were already working on.*

Also, how are data gaps in tower data and non-consistent temporal coverage of the towers treated? This might influence the analysis of the derived merging weights if the station sample changes over time.

*R. There are certainly gaps in our processed tower data because, even if a station data record was complete, we remove the rainy intervals. We deal with this by requiring a minimum number of 10 daily observations in the running window selecting the time interval to derive the daily weights. Otherwise the weights will not be computed for that specific day. The latter happens in a very few occasions so it is not critical for the results. Note that we changed the running window to 31 days as with the three models we noticed that this was more adequate than the original 61 days*

*Regarding temporal coverage, for the 84 stations considered, 6, 14, 24, 9, and 31 stations had 2, 3, 4, 5, and 6 years of data, respectively. Clearly, the shorter the time period the less confidence we should have in the weights because the weights may be too specific to the climate conditions of these few years (e.g., especially dry or wet conditions). If these weights were then applied to merge the products at that location but for more standard conditions, the merge may not be optimal.*

*So, yes, there can be an impact due to data gaps and temporal coverage, but how this is influencing the analysis is difficult to judge. A possibility to minimize the impact of the temporal coverage is to keep only towers having data for a relatively large number of years, but that considerably reduces the already small number of towers. For instance, allowing at least 4 years of data removes half of the towers considered in the study.*

Specific comments:

page 3, line 12: What about other observational data? E.g. lysimeters or catchment wide estimates could be mentioned as well here.

*R. We have used catchment wide estimates to evaluate our ET models during the WACMOS-ET project (Miralles et al., 2016), and we are indeed using them in the revised manuscript. But we would possibly not call them observations, at least not ground observations. Lysimeters are definitely ground observations. However, as far as we know there is no organized network of lysimeter measurements, which may facilitate their widespread use for global evaluation of ET models. Nevertheless, we can mention them. We will rephrase as: "Ground measurements of land heat fluxes are typically conducted during field experiments (Pauwels et al., 2008) and by more permanent lysimeters (Hirsch et al., 2017) and flux tower networks (Baldocchi et al., 2001)".*

page 5, lines 24/25: Do you expect an impact on the merging weighs when sub-daily simulations and tower data would be considered?

*R. Yes, but only if the ET models skills were markedly different at different times of the day. In that case the weights for specific times of the day would differ from the daily ones. However, from our sub-daily simulations presented in Michel et al. 2016 we did not notice very different skills at different times of the day when comparing with the tower data, and not much more skill was gained by producing daily ET based on 3-hourly input as opposed to forcing the models with the original daily input. We could then speculate that the weights will not be dramatically different in our case. In any case, we do not envisage to produce*

*more sub-daily runs within this project given that, on the one hand, we do not have had many requests for sub-daily simulations estimates, and on the other hand, ET model updates are mostly focusing only on the daily scale.*

page 7, line 18: Obsolete brackets between the two cited papers.

*R. Thanks, we will remove them.*

page 8, line 1: What happens if the two data sources for precipitation disagree?

*R. Certainly there are moments when the precipitation at the tower disagrees with the gridded precipitation, which is expected due to the different spatial resolutions and the unavoidable errors associated to the gridded product. We only leave days when there was no rain from the gridded product and the tower recordings (for the towers where precipitation was measured). We will rephrase to make it clearer as: "(2) masking measurements for rainy intervals, only leaving observations if both the global precipitation product and the local measurements (if available) do not indicate precipitation (eddy-covariance measurements are generally less reliable during precipitation events)".*

page 12, line 8, "estimated over the time series of available errors": What about differences in the length of the EC time series or differences in the occurrence of data gaps between the towers? How is this taken into account in the analysis of the weights?

*R. As we explained above, all the towers do not have the same number of years of data, and there are data gaps, especially because we filter for precipitating conditions. This could have an impact in the weights analysis, especially if for the stations with a very short number of years, the existing years are not representative of the typical climate conditions, but we cannot quantify the impact of this. Being much stricter with the data gaps and number of years was not possible here, as the number if towers and weights would have been considerably reduced, as previously discussed.*

*We will add the information about the number of years in the text and comment on the possible impact of a relatively short number of years to derive the weights. We will add: "Not all stations completely cover this period, with 6, 14, 24, 9, and 31 stations having 2, 3, 4, 5, and 6 years of data, respectively. At stations where inter-annual variability is large the weights may not be representative of the overall climate conditions at the tower if only a relatively*

*short number of years exist. Limiting the study to stations with a relatively large number of years could have been used to minimize the impact of this, but this severely reduced the number of towers, so this filtering has not been applied. For instance, if we only derive weights if at least 4 years of data are available, half of the towers would have been removed."*

page 13, lines 3-5: Rephrase: "A 61-day running window was found to provide ..."

*R. We will rephrase like that.*

page 13, lines 3-5: Is there a minimum requirement of data availability for deriving the weight within the window?

*R. Yes, 10 daily observations as mentioned before. We will add this to the text .*

page 13, lines 9-13: Sounds a bit confusing and not so clear (at this point of the paper at least). Try to re-formulate being a bit more specific (reasons to believe that?).

*R. We are just stating here the impossibility to evaluate interception as we mask the data record for precipitation conditions. We will rephrase trying to be clearer as: "GLEAM, PT-JPL, and PM-MOD estimate separately transpiration, soil evaporation, and the evaporation from the rain intercepted by canopies. Tower measurements will be masked for rainy intervals (see Section 3.2 }), so the interception loss of the modelled ET cannot be evaluated.  Therefore, only the sum of soil evaporation and transpiration is compared with the tower data and weighted. To derive the total ET merged product, an estimate of interception loss should also be provided, either by (1) assuming that GLEAM, PT-JPL, and PM-MOD interception loss are equally uncertain and adding their average to the weighted soil evaporation and transpiration, or; (2) by adding just one of the individual model interception losses, if there are reasons to believe that the selected one is less uncertain. Here we adopt the first approach, so the total ET product is the sum of the weighted soil evaporation and transpiration, together with the average of the three products interception losses".*

page 14, lines 14-20, "optimum product": How often is the optimum product one of the two models (i.e., weight of one)? From Fig. 5 it looks like a weight of one is never reached.

*R. Quite often, as there were many cases where both GLEAM and PT-JPL were*

*above or below the tower estimate. The situation is different now after the adding the third model, where in many occasions the optimum product is the tow er ET. For that reason, correlations of the optimum product with the tower get close to one at many stations, and the RMSD take low values. Therefore there are not much interest to define this target product and we remove the optimum product in the revised manuscript to simplify the discussion.*

page 19, Fig. 4: The legend interferes with the figure information, please increase the y-axis range a bit.

*R. Sure, we will fix that.*

page 19, Fig. 4 caption: Is it the optimum product you are comparing with the tower ET, not the WA-product? From page 14, I get that the optimum product is either the tower ET or one of the two models. Assuming that it's often the tower ET (judging from Fig. 5 where weights never reach one), shouldn't the RMSDs become zero? Perhaps, the relation of the optimum product and the WA-average product is not completely clear (at least to me) and might deserve some clarifications in the text.

*R. Yes, it was the optimum product. As we do not discuss the optimum product any more, we replace with the simple average product.*

Reviewer 3

This manuscript describes work to combine two ET products, PT-JPL and GLEAM, using a weighted average, with weight determined by fit to tower observations. The resulting product is limited to the locations of the towers, and no attempt is made at extrapolation to other sites. While the manuscript is well written, and the analysis sound, the work itself is not well motivated and, as currently presented, does not represent a significant contribution.

*R. We thank the reviewer for taking his/her time for a detailed review of our paper. As we explained in the public discussion, the local merging method was a first and necessary step towards testing a global merger. As we had already enough work on that direction, following the editor recommendation we have decided to revise largely the manuscript to include a global weights extrapolation.*

The merged product presented in this manuscript does not add any value to the ET products that are already available. The motivation seems to be to merge the two ET products (PT-JPL and GLEAM) to produce a new product that is as close to the tower ET time series as possible. How is this new merged product then any more useful than the original tower ET time series?

*R. The local merge of GLEAM and PT-JPL at the selected towers was the first step to produce a merge product. Perhaps we were not clear in the motivation and objectives. We fully agree with the reviewer that the merge product would be useful outside the locations of the towers, but not where we already have the tower estimates. But the first step is to prove that the merge product fits the tower data better than the individual products at the tower sites which is not an obvious exercise as shown in the paper.*

The merged ET product has not been independently evaluated. It is shown to be closer to the tower observations, but this is by design. Given that the tower observations also have errors (and given how closely the merged product has been fit to the tower obs), it does not follow that the merged product is necessarily more accurate. I am concerned that the merged product is over-fitting to the tower obs (weights calculated independently at each location, using a moving temporal window).

*R. For the moment we were just trying to show that at each specific site the optimal estimator could result in a product better fitting the tower ET than the original products. Now, for the global merger over-fitting definitively needs to be tested and we have incorporated some analysis now in the revised manuscript.*

*Certainly the tower ET also has errors, as described in the paper, and any methodology that tries to fit to the tower ET is likely to inherit those. Nevertheless, there is some consensus in the ET community that the tower fluxes are our best shot for ground "truth" at ecosystem scale. The optimal linear estimator applied here tries to minimize the error variance of the merged product with respect to a reference, in this case the tower observations, and in that sense certainly by definition the merge product tries to get closer to the tower observations, compared with the original ET estimates.*

The work is not very well motivated. Why merge just these two products? Why not merge as many as are available, or as many as meet some pre-defined standard? The selection of just these two products is particularly awkward given that they are not independent.

*R. We stated in the Introduction the reasons behind merging GLEAM and PT-JPL. In short, after years of testing different methodologies to derive "satellite-based" ET products, GLEAM and PT-JPL showed more skills than others tested methodologies (Michel et al., 2016, Miralles et al., 2016, McCabe et al., 2016), so we wanted to see if we could merge them to produce a better product. We think that this is a legitimate objective in the framework of our WACMOS-ET project and connected initiatives, such as the GEWEX LandFlux initiative (https://halo.kaust.edu.sa/Pages/GEWEX_Landflux.aspx), and we do not see anything awkward here even if the products are not completely independent. Nevertheless, we are adding the third model globally run by WACMOS-ET, the algorithm behind the MODIS MOD16 ET product, to have a more diverse palette of models.*

*We like to add here that for this type of EO-driven process-based daily ET products we are interested in, as far as we know there are no more alternatives right now apart from the WACMOS-ET and LandFlux runs we are producing, and that only one of them is publicly available with more independent forcings (GLEAM). Only when the teams from PT-JPL, PM-MOD, etc, produce daily estimates we may be in the position of testing again the merging with more diverse products in terms of forcings.*

To be publishable this work must i) provide a product that adds value in some way to the original products., and ii) the resulting data set must also be independently verified.

*R. We disagree that research on this topic can only be published if it results in a new product. We believe that what we learned about merging our products is of broad interest for other colleagues working on these topics, even if it is just a firs step for a successful merger.*

The most obvious way to achieve this would be to spatially extrapolate the weightings. This could potentially provide a new product with (near-) global coverage that is more accurate than either of the original gridded ET data sets, and would also allow independent verification against withheld tower observations.

R. *Extrapolating the weights is certainly required to produce a global merger and we have included that in the revised manuscript instead of leaving that for a second paper. The results of the weighted product do not clearly improve the simple-average product, but we still think that it is worth publishing the*

*outcome. In our humble view, there is a bit of "overselling" in current efforts to find adequate weights for a more informative merge than the simple average, so it is worth for us to publish our results. For instance, the recent https://doi.org/10.5194/hess-22-1317-2018 claims to produce a linear optimal combination of monthly ET products, but the relative improvements of the weighted product metrics (correlation, MSE, etc) with respect to the simple average product are nearly negligible (see Fig. 3, 4, etc).*

*We also have done the suggested tests of withholding one tower from the prediction data set and checking the prediction there. As you can see in the manuscript, these tests are showing that our tower data set is limited in terms of representing different biome and climate conditions, and a future merges needs to address these limitations.*

If this is not possible, I suggest that the manuscript be re-submitted and re-written (with additional discussion and conclusions) to focus on evaluating the GLEAM and PT-JPL products against tower obs.

*R. As discussed, we finally agreed to produce a global extrapolation of the weights, explore the representativeness of the tower data set, and present a global merge product, including a water catchment budget analysis. Still, we still consider this as an exploratory exercise and we are not claiming that we have succeeded in producing a great merged product. To make that clear, we are changing the title to "Exploring the merging of the global land evaporation WACMOS-ET products based on local tower measurements".*

MINOR COMMENTS:

Section 2: There is not enough information here for the reader to understand how the two products are calculated and what their main differences are. Please provide full details of the methodology of each product, rather than relying on previous work.

*R. This is the third paper of the WACMOS-ET project, the first two ones also published in this journal. The GLEAM, PT-JPL, and PM-MOD models were described in more detailed, including their main equations, in the first paper, while for the second and this third one we only describe the main characteristics of the models. We are certain that any reader interested in this work would need to glance through the previous papers to follow this one, so we are not sure that fully describing the models here will be that useful. The same applies to the*

*model forcings, which we described in detail in the first paper, and that we only summarized in the second and this third paper. We will consult with the editor about this, as we already had plagiarism complains precisely by mentioning again in this paper project elements already described in the first papers.*

P5, L24: give the specific resolutions.

*R. We will rephrase as: "Notice that the WACMOS-ET runs were done at 3-hourly and daily time resolutions, while only daily estimates are calculated for this study".*

P8, L5: mention that the station coverage is not globally uniform, with nearly all stations in Europe and the US.

*R. We will mention it.*

P8, L20: 'corrected fluxes are preferred". Provide citation. Also, for the results provided in this paragraph for the corrected fluxes, how were they corrected?

*R. Citations provided. Bowen ratio, the paragraph has been improved to add more details and better justify the use of the uncorrected fluxes.*

Equation 1: add a sentence to describe what this metric is measuring (something like "the first term is the mismatch between the land cover at the tower and at the grid cell level, and the remaining terms are the net mismatch in land cover types across the two resolutions").

*R. We will add as suggested: " …where the tower is situated. The first term is the mismatch between the land cover at the tower and at the grid cell level, and the remaining terms are the net mismatch in land cover types across the two resolutions. It takes the value …".*

P14, paragraph from line 10: the text here implies that the motivation is to match the tower ET as closely as possible, but the tower ET will also include errors. This paragraph should be re-written to acknowledge that the tower ET will also include errors (and the methodology perhaps adjusted to not over-fit to the ET data)

*R. We removed the optimum product to simplify the discussion. When we only had GLEAM and PT-JPL, both were above or below the tower estimate in many occasions, so the optimum product was one of the models. The situation is different now after adding the third model, where in many occasions the optimum product is the tower ET. For that reason, correlations of the optimum*

*product with the tower get close to one at many stations, and the RMSD take very low values. Therefore there are not much interest to define this target product and we remove the optimum product in the revised manuscript.*

P15, L10. The use of the full seasonal cycle concerns me. In general, different ET products agree reasonably well in terms of the seasonal cycle (Jimenez et al. 2011; Mueller et al. 2011; Miralles et al. 2011). It is the anomalies that have more disagreement, and should then be the focus of efforts to improve / combine ET products. Also, using anomalies would be consistent with the assumption in the methodology that there are no biases. The reason given for not using anomalies is that there is insufficient tower data - if there really is insufficient data, this implies that ET cannot be trained on tower obs.

Jimenez, C., and Coauthors, 2011: Global intercomparison of 12 land surface heat flux estimates. Journal of Geophysical Research: Atmospheres, 116, D02 102, doi:10.1029/2010JD014545. Miralles, D., T. Holmes, R. de Jeu, J. Gash, A. Meesters, and A. Dolman, 2011: Global land- surface evaporation estimated from satellite-based observations. Hydrology and Earth System Sciences, 15, 453–469, doi:10.5194/hess- 15-453-2011. Mueller, B., and Coauthors, 2011: Evaluation of global observations- based evapotranspiration datasets and IPCC AR4 simulations. Geophysical Research Letters, 38, L06 402, doi:10.1029/ 2010GL046230.

*R. The better agreement of ET products when the full seasonal cycle is considered is just the result of correlating two variables with marked lows and highs. In general, the more pronounced the seasonal cycle, the better the agreement in terms of correlation. At locations when the seasonal cycle is smaller, such as tropical forests, the agreement of the absolute ET values is much poorer in terms of correlation. This is not exclusive of ET estimates, it is also the case for other variable with strong seasonal cycles (e.g., radiation, temperature, precipitation).*

*Certainly, working with the anomalies would have been interesting, but this cannot be reliably achieved with our present data records. To work with anomalies, a robust calculation of the seasonal cycle at the tower locations is needed. How many years would be acceptable? If we take the whole 1980-2015 FLUXNET2015 synthesis data set, and we conserve stations having at least 10 years of data, we are left with ~25% of the stations. If we take 5 years, which can be disputed as a sufficient number of years for a climatology, we still*

*remove ~50% of the stations. The tower dataset is already severely limited in terms of geographical coverage, so such dramatic cuts in the number of stations is not very helpful for any merging methodology.*

*We do not agree with the reviewer comment that ET cannot be trained in tower observations if we cannot work with anomalies. It is still a big challenge to reproduce the absolute ET values, as clearly shown in the references given by the reviewer, or some figures in our manuscript, and as far as we can see most ET product merging efforts based on tower data work with absolute values.*

P18, L18: EC is known to under-estimate the fluxes. Using the sum of LH and SH as the incoming energy will almost certainly give an underestimate.

*R. True. We have been very clear about it in the same paragraph, stating the 6.1% underestimation when averaged over all the stations. We would argue than this underestimation has a smaller impact here, compared with other statistical approaches directly targeting the tower ET, such as the MTE product suggested later on by the reviewer. This is because we are not directly reproducing the tower ET, but weighting the original ET estimates. There can be an effect when deriving the error variance, as the relative differences of the original estimates with the tower ET can change if corrected or uncorrected tower fluxes are used, but the merge product still remains bound by the original estimates. To see if this effect was large, we recalculated the weights with corrected and uncorrected fluxes at the stations having both. We found good agreement between corrected and uncorrected weights, suggesting that the impact for this particular exercise is small.*

Figure 5: what is causing the sudden changes in the time series? The 91 day windows used shouldn't suddenly change like this.

*R. The reviewer is right, and continuous 61-day windows should not produce abrupt changes in this plot. However, at some locations the weights do not exist for all days. This happens at a few stations, as we imposed that there should be at least 20 well spread daily values in the 61-day running window to derive the weights. For instance, in the right panel of original Figure 5 the maximum values before the sudden decrease at day 180 correspond to the station CN-Dan. Due to observations quality and rain episodes there are not enough daily values to derive the weights for this station for the next few days, and the next maximum value comes from a new station with a lower value, producing the discontinuity.*

*We have changed the running window to 31 values, which seem more adequate for the new merge with three products. To describe the minimum number of values, we add to the text: "Due to the masking of the tower data at very few occasions the 31 daily estimates are present in the running window applied to derive the weights, and at least 10 daily values in the running window are required to derive a daily weight. Most stations have weights for nearly all days, but at 8 stations there are larger gaps. The worst case is the tropical BR-Sa3 station, where the frequent rainy episodes complicate the derivation of the weights". We also updated the figure to include the PM-MOD and replaced the minimum maximum weights with the 25% and 75% percentiles, which are less sensitive to the discontinuities caused by the gaps in the time series of the weights at a few station.*

Figure 7: This sudden increase in the tower ET in the upper panels look incorrect (and seem to occur at the same time each year - unless these are preceded by significant rain events, this don't look right). This time series needs to be checked, carefully QC- ed, and unusual features like this should be explained in the text.

*R. The tower data was quality-controlled using the provided quality flags, and the represented fluxes were not marked as problematic. This site is a semi-arid savannah where vegetation development and associated fluxes are tightly linked to precipitation and humidity conditions. Station precipitation and soil moisture measurements in the 2004-2007 period can be found in Scott et al., 2009 (J. Geophys. Res., 114, G04004, doi:10.1029/2008JG000900), and match the general behaviour of the fluxes.*

*However, we only plotted the ET estimates used for the tower-model agreement analyses, i.e, with the rainy episodes removed. This together with the running window used to smooth the lines produced the abrupt changes at the arrival of the summer rainfall, when many ET estimate are removed to derive the weights. This was very confusing, and the reviewer rightly spotted the problem. To remove any confusion, we will be re-plotting the full time series, not just the non-rainy days. Notice that we added a third station to the plot, replaced the Ca-Gro with a new station more illustrative for the merging of the 3 products, and shorten the time series to show only 2 years to help the readability of the plots.*

The work would benefit from being placed within the context of other efforts to estimate ET with tower data / statistical methods. In particular the MTE product should be mentioned somewhere, as an example of using tower EC obs to estimate global ET.

Jung, M., M. Reichstein, and A. Bondeau, 2009: Towards global empirical upscal- ing of FLUXNET eddy covariance observations: validation of a model tree ensemble approach using a biosphere model. Biogeosciences, 6, 2001–2013, doi:10.5194/bg-6- 2001-2009.

R. *We are very familiar with the MTE product, we compared the WACMOS-ET estimates with the MTE product in Miralles et al., 2016. This product is now cited as discussed above.*

[revised manuscript text omitted]

400 different for the first half of the year, while for the second part GLEAM is the most weighted product. At the US-SRM site, a semi-arid grassland site in North America, for the MAM period PM-MOD is much more weighted than GLEAM and PT-JPL, while for the other periods the weight differences are smaller. The last site, the US-Ne1 cropland station situated in North America , is an example of very close weights for all models, a situation that can be observed at quite some other stations.

[Figure]

**Figure 5.** Including PM-MOD weights and displaying only the weights at 3 stations Example of GLEAM (red), PT-JPL (blue), and PM-MOD (green) weights at the FR-Pue (top), US-SRM (middle), and US-Ne1 (bottom) stations.

**5.2 Merge products**

To illustrate the merged products, time series of the original and merged products for the three sites of Fig. 5 are plotted in Fig. 6. Only 2 years are displayed to help readability. The FR-Pue site shows large inter-annual variability related to the alternance of cold and warm seasons and the availability of soil moisture (Rambal et al., 2004). All products disagree with the tower ET in 2006 for a large part of the year, while in 2007 GLEAM agrees well with the tower. The largest weight for GLEAM helps getting the WA-merge product closer to the tower, compared with the SA-merge product. The US-SRM site also shows a relatively large ET seasonal variability, with the ET tightly linked to the precipitation and associated increased in soil moisture (Scott et al., 2009). GLEAM and PT-JPL capture better than PM-MOD the sudden increase in ET values at the beginning of summer related to the rainfall coming from the North American monsoon. The merged products capture well the summer ET rise, but fail to replicate the following largest ET values as all original ET estimates are below the tower ET. This is the consequence of the merged product values always being bound by the original ET estimates, and differs from other merging approaches where the ET tower is directly regress on a set of explanatory variables (Jung et al., 2011) or ET products (Yao et al., 2017). The differences between the SA-merge and WA-merge products is smaller than at FR-Pue, consequence of the closer weights at US-SRM during the months with larger ET. The US-Ne1 is a rainfed maizesoyabean irrigated site, with a expected more regular seasonal cycle and larger ET values than the two previous sites (Verma et al., 2005). The original products have have more similar values, not capturing well the ET rise associated with start of the growing season. The closer values results in closer weights and very close SA-merge and WA-merge products.

[Figure]

**Figure 6.** Updating figure to add PM-MOD and displaying shorter periods and one more station 2006-2007 
[revised manuscript text omitted]

---

## Author Response (AR2)

Dear Editor,

Please find below a point-by-point response to the reviews and a marked-up manuscript version. We have substantially changed the original manuscript to accommodate the demands from the second reviewer. Major changes are: (1) better explaining why we are limited to this sample of three ET products; (2) working with product anomalies to address the issues related to biases between the products; (3) implementing an inverse error-variance scheme to weight the anomalies, including the use of the full error covariance matrix to address the data sets dependencies; (4) better describing the NN methodology; and (5) more clearly stating whether for this particular exercise the weighted average can be considered advantageous compared with the simple average. Moreover, the readability has been improved by accommodating minor textual edits throughout the document. To guide the reviewers through the manuscript we have marked up specific request in red, clearly indicating the page and concerned lines in the response to the reviews, and adding short sentences in red in the manuscript to signal major changes to the text.

Sincerely yours,

Carlos Jimenez (on behalf of all co-authors)

**Reviewer 1**

**The authors have successfully addressed my comments on the previous version of the manuscript. The extension of the number of merged models as well as the extrapolation to the global scale improved the value of the analysis. As such, I only have minor comments/corrections at this stage. Note that the line numbers in the following refer to the file hess-2017-573-manuscript-version5.pdf.**

We thank the reviewer by going again through the manuscript and providing us with comments to improve it.

Specific comments:

**line 205: how was the cross validation performed (leave-one-out)?**

R: The cross-validation mentioned here is a standard technique to prevent the NN to over-fit to the training dataset, and it is applied here any time the NN is trained, no matter which training dataset is used. The leave-one-out is a different strategy, having the objective of assessing the quality of the dataset in terms of capturing the general distribution of the data. In

practical terms we test this by removing one station from the training data set, and checking if the weights of the station left aside can be properly replicated by the NN. If it is not the case, this is a strong indication pointing towards training dataset not robust enough to represent the all-conditions true distribution. These two options are now better discussed in P7-L209-227 of the revised manuscript, with the results of the tests presented in Section 7.3.

**line 212: remove one of the double "the"**

R: Corrected.

**line 218: wrong placement of brackets for Willmott (1982).**

R: Corrected.

**line 240: semicolon after "Release 3.1"**

R: Corrected

**line 321: "World" instead of "Wordl"**

R: Corrected

**line 392: delete "also" (the previous station showed large inter-annual variability, not seasonal variability)**

R: Corrected

**Figure    7    caption:    "axes"    instead    of    "axis"    in    the    last    sentence**

R: Corrected

**Figure 8: adjust the size of the lower right panel to match the other panels.**

R: Done.

**line 514: "first" instead of "firs"**

R: Corrected

**line 574: missing "and" in "0.93 (MSWEP) 0.88 (WorldClim)"**

R: Corrected

**line 577: "catchment water budget" sounds more commonly used to me (here and some other instances)?**

R: Replaced.

**line 631: "In" instead of "Tn"**

R: Corrected.

**line 634: "factor" instead of "factors"**

R: Corrected.

**Reviewer 2**

**The authors have expanded on their original submission, by extrapolating the estimated weights using a neural network, and adding a third product to the merger. While the manuscript is certainly improved from the first submission, I still have major reservations about the motivation for this work, and the appropriateness/accuracy of the chosen methodology. These issues would need to be address prior to publication.**

We thank the reviewer by going again through the manuscript and providing us with valuable comments to improve it. We have better motivated our work, updated the methodology to work with anomalies, and more clearly stated the conclusions of our exercise concerning the adequacy of using a weighted or unweigthed average for this particular combination of products.

**1.**
**The motivation for merging these particular products needs to be provided in the manuscript (not just in the reviewer response). Also, there are other products available that you could use, say from reanalyses.**

R: We are rephrasing this part of the introduction to more clearly state that this merging effort is within a specific ET modelling framework: the WACMOS-ET project. We are also mentioning that as far as we know there are no other daily global ET products publicly available. All this is added in P3-L58-68.

Concerning the reanalysis, their surface fluxes are a different type of product than the ones targeted here. They are normally classified as "derived" quantities, as their estimates are not directly constrained by the observations, and we do not intend to merge them here with the more observation-driven fluxes of the WACMOS-ET project. We have added an statement in the introduction to highlight this (P2-L43-48).

**Given that all three input ET products are forced with the same data, it is not clear to me why using a statistical method (based on random errors) would be beneficial to account for systematic differences (driven by model structure / parameters) between the three products. This is a fundamental flaw in this work.**

R: The three ET products share some common forcing data (namely the surface radiation and the air humidity), but they also use different products (e.g. the vegetation products and precipitation only being used by GLEAM). This is described in Section 3.1. Therefore, the differences are not only driven by model structure/parameters and the difference between the gridded products and the tower ET contains both random and systematic components, as

shown in the original Figure 11. Below, we discuss further why our merging methodology is adequate after updating it as suggested by the reviewer (see below).

**While this is not actually stated, it seems to me that the main point of this work is to determine whether these three products can be improved upon by merging them, and if so, can they be merged more effectively than using an (unweighted) average.**

R: We already stated in the introduction: "As such we pose the question: can a product combining the GLEAM, PT-JPL, and PM-MOD estimates result in a more accurate ET estimate? To make the paper objective clearer we are now also mentioning the simple average in the introduction, and that we are comparing it with the weighted product (P3-L76-79).

**There are two steps to this:**

**i. Are the weights (or more importantly, the final product) significantly different between the standard averaged, and with the eights calculated using a more sophisticated approach?**
**Currently this has not been adequately answered, due to the flawed method used to estimate the weights.**

R: See our comments below regarding our changes to the methodology.

**ii. Can the weights calculated at the limited number of tower locations be usefully extrapolated globally?**
**Currently this cannot be answered, as not enough information is given regarding the NN used to extrapolate the weights.**

R: See our comments below regarding the application of the NN.

**2. As in the first review. The inverse-error variance weights are based on the assumption of unbiased and independent data sets. The data sets used here are strongly dependent and biased, and this cannot be ignored. At a minimum for publication, the weights (and consequently the averaging) needs to be based on anomalies (to remove the bias, and the more systematic aspects of the dependence), and the lack of independence needs to be acknowledged prominently, including qualifying all conclusions by noting that these data sets were not independent. If you don't have enough data to use anomalies, then you cannot apply this method by ignoring the inconvenient biases.**

R: We fully agree with the reviewer that an inverse-error variance merging requires unbiased estimates. Only under those conditions this statistical method can be considered and optimal estimator, with the resulting merged product minimizing the error variance. Therefore, strictly speaking, if we consider the *in situ* observations unbiased with respect to the truth, the products should be debiased against the *in situ* observations. This kind of debiasing will strongly constrain the merged product to the tower absolute values, and would require a global estimation of the bias for the final product. In essence, this would be trying to reproduce the fluxes, as in Yao et al., 2017. Note that this is not the objective here, but the intention is to weight the original gridded products giving more weight to the products that reproduce the tower variations more closely.

After some tests we finally implemented a new merging scheme that allows working with anomalies, but without correcting to the tower absolute values. In short, we first derive time series of anomalies for each product and the tower observations by removing their individual seasonal climatologies. Then, we calculate the full covariance matrix of the difference of the gridded product anomalies and the tower anomalies as the basis for the weighting, so we account for the expected product correlations due to modelling/forcing similarities. Finally, we weight the anomalies and add them to the climatology of the simple-averaged product. This is now fully explained in P5-L155 to P6-L186.

**3. The NN network is not adequately explained, and reads like it has been blindly applied rather than carefully investigated. As with the first review, I still have concerns about over-fitting. Since so little information is given, it is not clear whether the NN was not useful for extrapolation, because of the limited locations of the tower observations, or because the NN itself was inadequate.**

R: We have ample experience applying a similar NN setup for different geophysical problems (e.g., Jiménez et al., 2003, Jiménez et al., 2009, Jiménez et al., 2012) and are certain that the NN is adequate for this application. The NN description was short as we firmly believe that the limitations here are coming from the data set, not from the method used to find the statistical relationship between the weights and the model inputs. Multi-layer perceptrons can be trained with different backpropagation algorithms, and their generalisation capacity can be optimised by techniques such as structural stabilisation, cross-validation, or regularisation. In our experience all these different NNs perform very similarly when properly set up. This has been further explained in the text (see below).

**For publication, the NN needs to be more adequately explained. This includes details of how the training data sets were split into training and evaluation data within the NN algorithm (and how this then relates to the experiments where a single station was removed), how the input data was selected / what else was tested, and how you ensure that the output weights equal 1.**

R: Full details are given now at P7-L197-227. It is worth indicating that controlling the optimal complexity of the model underlined by the NN is relatively easy by using some of the techniques mentioned before, so we are confident that we are not over-fitting in that sense. However, this does not tell much about the "quality" of the data set to truly sample all possible conditions, which will be always a concern when using the existing pool of tower data for global applications. Therefore, the typical stratification of the training data sets to test whether the available sampling of the true distribution is adequate to capture most possible conditions.

**Also, it isn't stated whether a separate NN is calculated for each day of the year, or if all data is thrown in to a single NN. I suspect it is the former, but would strongly recommend the latter.**

R: Yes, the NN is trained to model the annual statistical relationship between the weights and model inputs. We are making that clear now (P6-L191-193). We would not advice to model the daily relationship due to the resulting very small database for the individual daily trainings, and the suppression of the temporal variability in the daily training data sets. Both

things do not help define the link between variations in the NN inputs and weights, and are likely to impact the robustness of the found relationships.

**How do you deal with hemispheric differences? It doesn't really make sense to predict SH weights for a day of year, based on NH data only (there is only one SH tower, in Australia).**

R: We agree that this is a severe limitation for a global extrapolation. Still, we think it is worth presenting a global extrapolation, but testing if the biomes covered by the stations in the NH could be somehow representative of the conditions in the SH. This is a gross assumption, and our tests indeed showed that it is not the case. A test about this can be found in P25-L543 to P26-L560.

**By training a single NN using data for all days (and not including day of year in the training data) you could withhold data for some days to test whether the NN can at least reproduce the sites that are sampled.**

R: A similar test is now included in Section 7.3 and described in P25-L517-542. The "all stations" statistics are derived from days of the year not seen by the NN during the training phase. For that, from the original stations data set 15% of the available days at each station are removed before training and used for the statistics.

MINOR:

**1A general grammar check would be useful, with attention paid to missing possessive apostrophes.**

R: The whole manuscript has been carefully edited paying attention to the grammar.

**L49: please double check that the MTE product is a regression. I thought it was a machine learning approach, but could be wrong.**

R: Regression predictive modelling, i.e. is the task of approximating a mapping function (f) from input variables (X) to a continuous output variable (y), is a standard technique in machine learning approaches. Machine learning approaches are typically applied to either regression or classification problems.

**Section 2.1 (from first review also). Add the spatial resolutions in here.**

R: Added.

**Replace all instances of 'inverse-variance weighting' with 'inverse error variance weighting'**

R: Corrected.

**L210. "Only the systematic component of the error can potentially be captured by the NN, and there is not warranty that all the systematic errors are dependent on the model inputs". This is incorrect/misleading. "systematic errors" usually refers to biases, and**

**these methods can predict the mean square of the random errors (which is what your inverse error variance weights should reflect). Note that statistical methods like this do not predict individual errors, rather they predict the tendency towards larger/smaller errors.**

R: Yes, the sentence is indeed misleading, and we have rephrased it. Apart from that, we assume that by statistical methods the reviewer refers to the NN. If that is the case, we are certainly not trying to predict individual errors, but to model the distribution of the weights, conditioned to the ET model inputs and model ET estimates. For this, the NN is calibrated by minimizing the sum of square errors, where each error is the difference between each target weight of the training data set, and the NN prediction.

**Regarding the second half of the sentence, this is why you need to do some work to show that you have selected appropriate input data sets for NN.**

R: We agree that this part of the sentence was again not clear. We meant that the NN is used to model the statistical distribution of the weights, with this distribution not only depending on the variables used as predictors in the NN approach, which means that we can never perfectly predict the weights. This has been added to the manuscript (P7-L194-197).

**L273: quantify 'too close'. How is "clearly not representing the overall land cover" determined?**

R: We agree with the reviewer that this is rather subjective. In this context, "too close" meant that a water body could be visually identified in an aerial picture of the 25 km cell surrounding the eddy-covariance tower. The same visual inspection was used to discard stations where the station surroundings were clearly not representative of the station cover. This has been added to the manuscript (P10-L285).

**Explicitly note here that the towers do not provide comprehensive global coverage, and don't cover many biomes, climate regimes, or the Southern Hemisphere.**

R: We believe that this was already stated in the paper, but we are further adding that there are only 2 stations in the Souther Hemisphere (P11-L290)

**L280: representativity errors should be mentioned here (tower to 25 km).**

R: Certainly, representativeness errors cannot be ignored given the large mismatch between tower fetch and the spatial support of satellite imagery, but this section discusses the tower errors independent of the application. We chose to place this comment at the start of the next section, when we introduce some ancillary data to investigate the tower surroundings homogeneity (P13-L331-334).

**Figure 2 : 100 Wm2 is a huge range for each colorbar. Please plot with a finer discretization.**

R: 100 W/m2 corresponds here to a discretization of the full ET range in 10 intervals and the corresponding 10 colours. In our view this is an adequate level of discretization for this type

of global plots, and we have already used it in other flux related publications without any issues (e.g., Jiménez et al., 2009, Jiménez et al., 2010). Therefore, we prefer to leave the plot as such.

**Figure 4: It is stated repeatedly that the weights don't differ much from 1/3, and yet this plot shows a large deviation. Statistics are needed here to quantify the divergence from 1/3.**

R: The weights differ now from the 1/3 value after the new merging scheme. Nevertheless, we are replacing Figure 4 with a box plot to display the weight statistics in a more informative way.

**Figure 7: It doesn't really make sense to plot global fields for three month blocks.**

R: The authors do not fully agree with this comment, as these maps show the average seasonal patterns of evaporation, which helps to assess whether the temporal dynamics in the datasets are well represented at the seasonal scale.

**Figrue 10: This plot is difficult to understand. Why not plot Ih v. RMSE?**

R: We thank the reviewer for this suggestion. A scatter plot of Ih versus RMSD is certainly also a possibility, but here we intended to show whether a decreasing homogeneity results in an increasing RMSD. Plotting like this we can also display the linear square fit of the normalized RMSD of the sorted stations, so we prefer this type of figure. We are rewriting the figure caption to make this clear.

**Ih has not been defined anywhere in the manuscript.**

R: We would like to point the reviewer to Eq. 7 of the original manuscript, where Ih was already defined.

**L507: what about systematic errors in the obs? (inc. representativity)?**

R: This is indeed an issue, and it can be problematic when the observations are used to improve model estimates. Note that possible systematic errors in the tower observations were already discussed in Section 3.2 of the original manuscript though , now at (P11-L292-299).

**L509: "if the difference with the observations were mostly random in nature, we should not expect the observations to provide much guidance to combined the products". This is not correct. See comment on L210.**

R: We agree with the reviewer. Only in case of using the observations to improve a particular model estimate, the systematic differences are useful to detect and correct model issues. In the framework of an inverse error variance scheme to combine estimates the variance of the random errors guides the weighting.

**L521: No. You could have defined the bias over the data time period that you do have.**

R: We agree with the reviewer that the bias is an issue, and we believe we properly address this now by working on anomaly time series.

**L565: the occurrence of negative weights is because you have dependent products.**

R: This is indeed the underlying reason, and this is clearly acknowledged now throughout the manuscript (e.g. P17-L400).

**L581: Qualify that this approach assumes that the surface water storage has not changed over this time period, and that this assumption will not necessarily hold over the limitted time period you are using.**

R: This has now been acknowledged at P27-L577-578.

**Note that this is the same precip used in the ET products, and that this does not represent an independent evaluation.**

R: We agree, and we would like to point out that we were already tackling this issue by adding a second precipitation product. This is now highlighted again in Section 7.4 (P27-L570-575).

**L605: either explain why the slope is an important statistic here, or delete it.**

R: Given that the expectation here is the ET from the models equalling the basin ET inferred, closeness of the slope of the linear fit to one is a desirable target. Nevertheless, with the new merging scheme the slopes of the merged product are more similar, so we remove the slope discussion and add the bias as a third statistic inference.

[revised manuscript text omitted]

---

## Author Response (AR3)

Dear Editor,

Please find below a point-by-point response to the technical corrections suggested by the reviewer and a marked-up manuscript version.

Sincerely yours,

[Figure]

Carlos Jimenez (on behalf of all co-authors)

**Reviewer 1**

We thank the reviewer by going again through the manuscript and providing us with some technical corrections.

Specific comments:

**Abstract L15: fix grammar/typo.**

R: Resulted changed to remains.

**L66: qualify this statement with something like "satellite driven" or "purpose designed". Reanalyses (e.g., ERA5) also provide global fluxes at this resolution.**

R: Changed to "global satellite-driven ET estimates".

**equation 1: fix inconsistency between Ecm and Emc on line 159**

R: Corrected to Ecm.

**L171: Add a sentence here to state that C is estimated by comparison to the tower observations.**

R: Changed to "estimated by comparison to the tower observations, i.e., from the differences Eam – EaO".

**L432; "better estimates of ET". Rephrase to something like "better fit with the tower ET". WA-merger is closer to the tower observations by construction. Since you are not evaluating against independent observations, you can't conclude that this is necessarily better.**

R: Changed to "better fit with the tower ET".

**Section 6.1. Acknowledge in here that the southern hemisphere estimates are mostly based on northern hemisphere data, so there is a mismatch between the local seasons.**

R: We are not sure we understand this remark. It is true that the regression between weights and ET drivers and estimates contains mostly northern hemisphere stations. Therefore the regression can be considered northern hemisphere biased, as already explained in the manuscript. However, when applied to predict the weights the right seasonal predictors are used. For instance, for a boreal summer day and two locations situated in different hemispheres, the northern location predictors will contain northern summer values resulting in typical summer weights, while the southern location prediction will contain southern winter values producing typical winter weights. We do not see a seasonal mismatch in that sense.

**The content under the "Discussion" heading is more results. Delete this heading.**

R: We remove the Discussion heading and replace it with "7. Considerations on the merging" grouping the original 7.1, 7.2, and 7.3 discussions about the merging, and have a new independent heading for the evaluation "8. Merge products evaluation".

**Sentence end page 22 / start 23. Rephrase this. It is far more common to address biases in this situation by changing the variable to anomalies, which are unbiased (have zero-mean) by construction.**

R : Changing the variable to anomalies certainly allows the optimal merge of the anomalies, as we have done in the paper, but it does not solve the problem of optimally merging the absolute value of the estimates. This is the typical goal for many merged products, including our merging. Only by removing the bias between the original estimates and the observations can the absolute estimates be optimally merged in the sense described in the paper. We appreciate the comment, but we think that our sentence is appropriate in this context, and we leave it as it is.

[revised manuscript text omitted]